# NESTT: A Nonconvex Primal-Dual Splitting Method for Distributed and Stochastic Optimization

**Davood Hajinezhad, Mingyi Hong** [*]        **Tuo Zhao**[†]        **Zhaoran Wang**[‡]

## Abstract

We study a stochastic and distributed algorithm for nonconvex problems whose objective consists of a sum of $N$ nonconvex $L_i/N$-smooth functions, plus a nonsmooth regularizer. The proposed NonconvEx primal-dual SpliTTing (NESTT) algorithm splits the problem into $N$ subproblems, and utilizes an augmented Lagrangian based primal-dual scheme to solve it in a distributed and stochastic manner. With a special non-uniform sampling, a version of NESTT achieves $\epsilon$-stationary solution using $\mathcal{O}((\sum_{i=1}^{N} \sqrt{L_i/N})^2/\epsilon)$ gradient evaluations, which can be up to $\mathcal{O}(N)$ times better than the (proximal) gradient descent methods. It also achieves Q-linear convergence rate for nonconvex $\ell_1$ penalized quadratic problems with polyhedral constraints. Further, we reveal a fundamental connection between *primal-dual* based methods and a few *primal only* methods such as IAG/SAG/SAGA.

## 1  Introduction

Consider the following nonconvex and nonsmooth constrained optimization problem

$$\min_{z \in Z} \quad f(z) := \frac{1}{N} \sum_{i=1}^{N} g_i(z) + g_0(z) + p(z), \tag{1.1}$$

where $Z \subseteq \mathbb{R}^d$; for each $i \in \{0, \cdots, N\}$, $g_i : \mathbb{R}^d \to \mathbb{R}$ is a smooth possibly nonconvex function which has $L_i$-Lipschitz continuous gradient; $p(z) : \mathbb{R}^d \to \mathbb{R}$ is a lower semi-continuous convex but possibly nonsmooth function. Define $g(z) := \frac{1}{N} \sum_{i=1}^{N} g_i(z)$ for notational simplicity.

Problem (1.1) is quite general. It arises frequently in applications such as machine learning and signal processing; see a recent survey [7]. In particular, each smooth functions $\{g_i\}_{i=1}^{N}$ can represent: 1) a mini-batch of loss functions modeling data fidelity, such as the $\ell_2$ loss, the logistic loss, etc; 2) nonconvex activation functions for neural networks, such as the logit or the tanh functions; 3) nonconvex utility functions used in signal processing and resource allocation, see [4]. The smooth function $g_0$ can represent smooth nonconvex regularizers such as the non-quadratic penalties [2], or the smooth part of the SCAD or MCP regularizers (which is a concave function) [26]. The convex function $p$ can take the following form: 1) nonsmooth convex regularizers such as $\ell_1$ and $\ell_2$ functions; 2) an indicator function for convex and closed feasible set $Z$, denoted as $\iota_Z(\cdot)$; 3) convex functions without global Lipschitz continuous gradient, such as $p(z) = z^4$ or $p(z) = 1/z + \iota_{z \geq 0}(z)$.

In this work we solve (1.1) in a stochastic and distributed manner. We consider the setting in which $N$ distributed agents each having the knowledge of one smooth function $\{g_i\}_{i=1}^{N}$, and they are connected to a cluster center which handles $g_0$ and $p$. At any given time, a randomly selected agent is activated and performs computation to optimize its local objective. Such distributed computation model has been popular in large-scale machine learning and signal processing [6]. Such model is also closely related to the (centralized) stochastic *finite-sum* optimization problem [1, 9, 14, 15,

---
[*]Department of Industrial & Manufacturing Systems Engineering and Department of Electrical & Computer Engineering, Iowa State University, Ames, IA, {dhaji,mingyi}@iastate.edu
[†]School of Industrial and Systems Engineering, Georgia Institute of Technology tourzhao@gatech.edu
[‡]Department of Operations Research, Princeton University,zhaoran@princeton.edu

21, 22], in which each time the iterate is updated based on the gradient information of a random component function. One of the key differences between these two problem types is that in the distributed setting there can be disagreement between local copies of the optimization variable $z$, while in the centralized setting only one copy of $z$ is maintained.

**Our Contributions.** We propose a class of NonconvEx primal-dual SpliTTing (NESTT) algorithms for problem (1.1). We split $z \in \mathbb{R}^d$ into local copies of $x_i \in \mathbb{R}^d$, while enforcing the equality constraints $x_i = z$ for all $i$. That is, we consider the following reformulation of (1.1)

$$\min_{x, z \in \mathbb{R}^d} \quad \ell(x, z) := \frac{1}{N} \sum_{i=1}^N g_i(x_i) + g_0(z) + h(z), \quad \text{s.t. } x_i = z, \ i = 1, \cdots, N, \qquad (1.2)$$

where $h(z) := \iota_Z(z) + p(z)$, $x := [x_1; \cdots; x_N]$. Our algorithm uses the Lagrangian relaxation of the equality constraints, and at each iteration a (possibly non-uniformly) randomly selected primal variable is optimized, followed by an approximate dual ascent step. Note that such splitting scheme has been popular in the convex setting [6], but not so when the problem becomes nonconvex.

The NESTT is one of the first stochastic algorithms for distributed nonconvex nonsmooth optimization, with provable and nontrivial convergence rates. Our main contribution is given below. First, in terms of some primal and dual optimality gaps, NESTT converges sublinearly to a point belongs to stationary solution set of (1.2). Second, NESTT converges Q-linearly for certain nonconvex $\ell_1$ penalized quadratic problems. To the best of our knowledge, this is the first time that linear convergence is established for stochastic and distributed optimization of such type of problems. Third, we show that a gradient-based NESTT with *non-uniform sampling* achieves an $\epsilon$-stationary solution of (1.1) using $\mathcal{O}((\sum_{i=1}^N \sqrt{L_i/N})^2/\epsilon)$ gradient evaluations. Compared with the classical gradient descent, which in the worst case requires $\mathcal{O}(\sum_{i=1}^N L_i/\epsilon)$ gradient evaluation to achieve $\epsilon$-stationarity, our obtained rate can be up to $\mathcal{O}(N)$ times better in the case where the $L_i$'s are not equal.

Our work also reveals a fundamental connection between *primal-dual* based algorithms and the *primal only* average-gradient based algorithm such as SAGA/SAG/IAG [5, 9, 22]. With the key observation that the dual variables in NESTT serve as the "memory" of the past gradients, one can specialize NESTT to SAGA/SAG/IAG. Therefore, NESTT naturally generalizes these algorithms to the nonconvex nonsmooth setting. It is our hope that by bridging the primal-dual splitting algorithms and primal-only algorithms (in *both* the convex and nonconvex setting), there can be significant further research developments benefiting both algorithm classes.

**Related Work.** Many stochastic algorithms have been designed for (1.2) when it is convex. In these algorithms the component functions $g_i$'s are randomly sampled and optimized. Popular algorithms include the SAG/SAGA [9, 22], the SDCA [23], the SVRG [14], the RPDG [15] and so on. When the problem becomes nonconvex, the well-known incremental based algorithm can be used [3, 24], but these methods generally lack convergence rate guarantees. The SGD based method has been studied in [10], with $\mathcal{O}(1/\epsilon^2)$ convergence rate. Recent works [1] and [21] develop algorithms based on SVRG and SAGA for a special case of (1.1) where the entire problem is smooth and unconstrained. To the best of our knowledge there has been no stochastic algorithms with provable, and non-trivial, convergence rate guarantees for solving problem (1.1). On the other hand, distributed stochastic algorithms for solving problem (1.1) in the nonconvex setting has been proposed in [13], in which each time a randomly picked subset of agents update their local variables. However there has been no convergence rate analysis for such distributed stochastic scheme. There has been some recent distributed algorithms designed for (1.1) [17], but again without global convergence rate guarantee.

**Preliminaries.** The augmented Lagrangian function for problem (1.1) is given by:

$$L(x, z; \lambda) = \sum_{i=1}^N \left( \frac{1}{N} g_i(x_i) + \langle \lambda_i, x_i - z \rangle + \frac{\eta_i}{2} \|x_i - z\|^2 \right) + g_0(z) + h(z), \qquad (1.3)$$

where $\lambda := \{\lambda_i\}_{i=1}^N$ is the set of dual variables, and $\eta := \{\eta_i > 0\}_{i=1}^N$ are penalty parameters.

We make the following assumptions about problem (1.1) and the function (1.3).

A-(a) The function $f(z)$ is bounded from below over $Z \cap \text{int}(\text{dom } f)$: $\underline{f} := \min_{z \in Z} f(z) > -\infty$. $p(z)$ is a convex lower semi-continuous function; $Z$ is a closed convex set.

A-(b) The $g_i$'s and $g$ have Lipschitz continuous gradients, i.e.,

$$\|\nabla g(y) - \nabla g(z)\| \leq L\|y - z\|, \text{and} \quad \|\nabla g_i(y) - \nabla g_i(z)\| \leq L_i\|y - z\|, \ \forall \, y, z$$

---

**Algorithm 1** NESTT-G Algorithm

---

1: **for** $r = 1$ **to** $R$ **do**
2:      Pick $i_r \in \{1, 2, \cdots, N\}$ with probability $p_{i_r}$ and update $(x, \lambda)$

$$x_{i_r}^{r+1} = \arg\min_{x_{i_r}} V_{i_r}\left(x_{i_r}, z^r, \lambda_{i_r}^r\right); \tag{2.4}$$

$$\lambda_{i_r}^{r+1} = \lambda_{i_r}^r + \alpha_{i_r} \eta_{i_r}\left(x_{i_r}^{r+1} - z^r\right); \tag{2.5}$$

$$\lambda_j^{r+1} = \lambda_j^r, \quad x_j^{r+1} = z^r, \quad \forall\, j \neq i_r; \tag{2.6}$$

        Update $z$:     $z^{r+1} = \arg\min_{z \in Z} L(\{x_i^{r+1}\}, z; \lambda^r). \tag{2.7}$

3: **end for**
4: **Output:** $(z^m, x^m, \lambda^m)$ where $m$ randomly picked from $\{1, 2, \cdots, R\}$.

---

Clearly $L \leq 1/N \sum_{i=1}^N L_i$, and the equality can be achieved in the worst case. For simplicity of analysis we will further assume that $L_0 \leq \frac{1}{N}\sum_{i=1}^N L_i$.

A-(c) Each $\eta_i$ in (1.3) satisfies $\eta_i > L_i/N$; if $g_0$ is nonconvex, then $\sum_{i=1}^N \eta_i > 3L_0$.
Assumption A-(c) implies that $L(x, z; \lambda)$ is *strongly convex* w.r.t. each $x_i$ and $z$, with modulus $\gamma_i := \eta_i - L_i/N$ and $\gamma_z = \sum_{i=1}^N \eta_i - L_0$, respectively [27, Theorem 2.1].

We then define the *prox-gradient* (pGRAD) for (1.1), which will serve as a measure of stationarity. It can be checked that the pGRAD vanishes at the set of stationary solutions of (1.1) [20].

**Definition 1.1.** *The proximal gradient of problem* (1.1) *is given by (for any $\gamma > 0$)*

$$\tilde{\nabla} f_\gamma(z) := \gamma\left(z - \text{prox}_{p+\iota_Z}^\gamma[z - 1/\gamma \nabla(g(z) + g_0(z))]\right), \; with \; \text{prox}_{p+\iota_Z}^\gamma[u] := \underset{u \in Z}{\text{argmin}}\; p(u) + \frac{\gamma}{2}\|z - u\|^2.$$

## 2 The NESTT-G Algorithm

**Algorithm Description.** We present a primal-dual splitting scheme for the reformulated problem (1.2). The algorithm is referred to as the NESTT with Gradient step (NESTT-G) since each agent only requires to know the gradient of each component function. To proceed, let us define the following function (for some constants $\{\alpha_i > 0\}_{i=1}^N$):

$$V_i(x_i, z; \lambda_i) = \frac{1}{N} g_i(z) + \frac{1}{N}\langle \nabla g_i(z), x_i - z \rangle + \langle \lambda_i, x_i - z \rangle + \frac{\alpha_i \eta_i}{2}\|x_i - z\|^2.$$

Note that $V_i(\cdot)$ is related to $L(\cdot)$ in the following way: it is a quadratic approximation (approximated at the point $z$) of $L(x, y; \lambda)$ w.r.t. $x_i$. The parameters $\alpha := \{\alpha_i\}_{i=1}^N$ give some freedom to the algorithm design, and they are critical in improving convergence rates as well as in establishing connection between NESTT-G with a few primal only stochastic optimization schemes.

The algorithm proceeds as follows. Before each iteration begins the cluster center broadcasts $z$ to everyone. At iteration $r + 1$ a randomly selected agent $i_r \in \{1, 2, \cdots N\}$ is picked, who minimizes $V_{i_r}(\cdot)$ w.r.t. its local variable $x_{i_r}$, followed by a dual ascent step for $\lambda_{i_r}$. The rest of the agents update their local variables by simply setting them to $z$. The cluster center then minimizes $L(x, z; \lambda)$ with respect to $z$. See Algorithm 1 for details. We remark that NESTT-G is related to the popular ADMM method for *convex* optimization [6]. However our particular update schedule (randomly picking $(x_i, \lambda_i)$ plus deterministic updating $z$), combined with the special $x$-step (minimizing an approximation of $L(\cdot)$ evaluated at a different block variable $z$) is not known before. These features are critical in our following rate analysis.

**Convergence Analysis.** To proceed, let us define $r(j)$ as the last iteration in which the $j$th block is picked before iteration $r + 1$. i.e. $r(j) := \max\{t \mid t < r + 1, j = i(t)\}$. Define $y_j^r := z^{r(j)}$ if $j \neq i_r$, and $y_{i_r}^r = z^r$. Define the filtration $\mathcal{F}^r$ as the $\sigma$-field generated by $\{i(t)\}_{t=1}^{r-1}$.

A few important observations are in order. Combining the $(x, z)$ updates (2.4) – (2.7), we have

$$x_q^{r+1} = z^r - \frac{1}{\alpha_q \eta_q}(\lambda_q^r + \frac{1}{N}\nabla g_q(z^r)), \; \frac{1}{N}\nabla g_q(z^r) + \lambda_q^r + \alpha_q \eta_q(x_q^{r+1} - z^r) = 0, \; with \; q = i_r \tag{2.8a}$$

$$\lambda_{i_r}^{r+1} = -\frac{1}{N}\nabla g_{i_r}(z^r), \; \lambda_j^{r+1} = -\frac{1}{N}\nabla g_j(z^{r(j)}), \; \forall\, j \neq i_r, \; \Rightarrow \lambda_i^{r+1} = -\frac{1}{N}\nabla g_i(y_i^r), \; \forall\, i \tag{2.8b}$$

$$x_j^{r+1} \overset{(2.6)}{=} z^r \overset{(2.8b)}{=} z^r - \frac{1}{\alpha_j \eta_j}(\lambda_j^r + \frac{1}{N}\nabla g_j(z^{r(j)})), \; \forall\, j \neq i_r. \tag{2.8c}$$

The key here is that the dual variables serve as the "memory" for the past gradients of $g_i$'s. To proceed, we first construct a *potential function* using an *upper bound* of $L(x, y; \lambda)$. Note that

$$\frac{1}{N}g_j(x_j^{r+1}) + \langle \lambda_j^r, x_j^{r+1} - z^r \rangle + \frac{\eta_j}{2}\|x_j^{r+1} - z^r\|^2 = \frac{1}{N}g_j(z^r), \; \forall\, j \neq i_r \tag{2.9}$$

$$\frac{1}{N}g_{i_r}(x_{i_r}^{r+1}) + \langle \lambda_{i_r}^r, x_{i_r}^{r+1} - z^r \rangle + \frac{\eta_i}{2}\|x_{i_r}^{r+1} - z^r\|^2$$

$$\overset{(i)}{\leq} \frac{1}{N}g_{i_r}(z^r) + \frac{\eta_{i_r} + L_{i_r}/N}{2}\|x_{i_r}^{r+1} - z^r\|^2$$

$$\overset{(ii)}{=} \frac{1}{N}g_{i_r}(z^r) + \frac{\eta_{i_r} + L_{i_r}/N}{2(\alpha_{i_r}\eta_{i_r})^2}\|1/N(\nabla g_{i_r}(y_{i_r}^{r-1}) - \nabla g_{i_r}(z^r))\|^2 \tag{2.10}$$

where (i) uses (2.8b) and applies the descent lemma on the function $1/Ng_i(\cdot)$; in (ii) we have used (2.5) and (2.8b). Since each $i$ is picked with probability $p_i$, we have

$$\mathbb{E}_{i_r}[L(x^{r+1}, z^r; \lambda^r) \mid \mathcal{F}^r]$$

$$\leq \sum_{i=1}^N \frac{1}{N}g_i(z^r) + \sum_{i=1}^N \frac{p_i(\eta_i + L_i/N)}{2(\alpha_i\eta_i)^2}\|1/N(\nabla g_i(y_i^{r-1}) - \nabla g_i(z^r))\|^2 + g_0(z^r) + h(z^r)$$

$$\leq \sum_{i=1}^N \frac{1}{N}g_i(z^r) + \sum_{i=1}^N \frac{3p_i\eta_i}{(\alpha_i\eta_i)^2}\|1/N(\nabla g_i(y_i^{r-1}) - \nabla g_i(z^r))\|^2 + g_0(z^r) + h(z^r) := Q^r,$$

where in the last inequality we have used Assumption [A-(c)]. In the following, we will use $\mathbb{E}_{\mathcal{F}^r}[Q^r]$ as the potential function, and show that it decreases at each iteration.

**Lemma 2.1.** *Suppose Assumption A holds, and pick*

$$\alpha_i = p_i = \beta\eta_i, \; \text{where } \beta := \frac{1}{\sum_{i=1}^N \eta_i}, \quad \text{and} \quad \eta_i \geq \frac{9L_i}{Np_i}, \quad i = 1, \cdots N. \tag{2.11}$$

*Then the following descent estimate holds true for NESTT-G*

$$\mathbb{E}[Q^r - Q^{r-1}|\mathcal{F}^{r-1}] \leq -\frac{\sum_{i=1}^N \eta_i}{8}\mathbb{E}_{z^r}\|z^r - z^{r-1}\|^2 - \sum_{i=1}^N \frac{1}{2\eta_i}\|\frac{1}{N}(\nabla g_i(z^{r-1}) - \nabla g_i(y_i^{r-2}))\|^2. \tag{2.12}$$

**Sublinear Convergence.** Define the optimality gap as the following:

$$\mathbb{E}[G^r] := \mathbb{E}\left[\|\tilde{\nabla}_{1/\beta}f(z^r)\|^2\right] = \frac{1}{\beta^2}\mathbb{E}\left[\|z^r - \text{prox}_h^{1/\beta}[z^r - \beta\nabla(g(z^r) + g_0(z^r))]\|^2\right]. \tag{2.13}$$

Note that when $h, g_0 \equiv 0$, $\mathbb{E}[G^r]$ reduces to $E[\|\nabla g(z^r)\|^2]$. We have the following result.

**Theorem 2.1.** *Suppose Assumption A holds, and pick (for $i = 1, \cdots, N$)*

$$\alpha_i = p_i = \frac{\sqrt{L_i/N}}{\sum_{i=1}^N \sqrt{L_i/N}}, \; \eta_i = 3\left(\sum_{i=1}^N \sqrt{L_i/N}\right)\sqrt{L_i/N}, \; \beta = \frac{1}{3(\sum_{i=1}^N \sqrt{L_i/N})^2}. \tag{2.14}$$

*Then every limit point generated by NESTT-G is a stationary solution of problem (1.2). Further,*

$$1) \; \mathbb{E}[G^m] \leq \frac{80}{3}\left(\sum_{i=1}^N \sqrt{L_i/N}\right)^2 \frac{\mathbb{E}[Q^1 - Q^{R+1}]}{R};$$

$$2) \; \mathbb{E}[G^m] + \mathbb{E}\left[\sum_{i=1}^N 3\eta_i^2\|x_i^m - z^{m-1}\|^2\right] \leq \frac{80}{3}\left(\sum_{i=1}^N \sqrt{L_i/N}\right)^2 \frac{\mathbb{E}[Q^1 - Q^{R+1}]}{R}.$$

Note that Part (1) is useful in the *centralized* finite-sum minimization setting, as it shows the sublinear convergence of NESTT-G, measured only by the primal optimality gap evaluated at $z^r$. Meanwhile, part (2) is useful in the *distributed* setting, as it also shows that the expected constraint violation, which measures the consensus among agents, shrinks in the same order. We also comment that the above result suggests that to achieve an $\epsilon$-stationary solution, the NESTT-G requires about $\mathcal{O}\left(\left(\sum_{i=1}^N \sqrt{L_i/N}\right)^2/\epsilon\right)$ number of gradient evaluations (for simplicity we have ignored an additive $N$ factor for evaluating the gradient of the entire function at the initial step of the algorithm).

---
**Algorithm 2** NESTT-E Algorithm
---
1: **for** $r = 1$ **to** $R$ **do**
2:   Update $z$ by minimizing the augmented Lagrangian:

$$z^{r+1} = \arg\min_z \ L(x^r, z; \lambda^r). \tag{3.15}$$

3:   Randomly pick $i_r \in \{1, 2, \cdots N\}$ with probability $p_{i_r}$:

$$x_{i_r}^{r+1} = \operatorname*{argmin}_{x_{i_r}} U_{i_r}(x_{i_r}, z^{r+1}; \lambda_{i_r}^r); \tag{3.16}$$

$$\lambda_{i_r}^{r+1} = \lambda_{i_r}^r + \alpha_{i_r}\eta_{i_r}\left(x_{i_r}^{r+1} - z^{r+1}\right); \tag{3.17}$$

$$x_j^{r+1} = x_j^r, \quad \lambda_j^{r+1} = \lambda_j^r \quad \forall \, j \neq i_r. \tag{3.18}$$

4: **end for**
5: **Output:** $(z^m, x^m, \lambda^m)$ where $m$ randomly picked from $\{1, 2, \cdots, R\}$.
---

It is interesting to observe that our choice of $p_i$ is proportional to the *square root* of the Lipschitz constant of each component function, rather than to $L_i$. Because of such choice of the sampling probability, the derived convergence rate has a mild dependency on $N$ and $L_i$'s. Compared with the conventional gradient-based methods, our scaling can be up to $N$ times better. Detailed discussion and comparison will be given in Section 4.

Note that similar sublinear convergence rates can be obtained for the case $\alpha_i = 1$ for all $i$ (with different scaling constants). However due to space limitation, we will not present those results here.

**Linear Convergence.** In this section we show that the NESTT-G is capable of linear convergence for a family of nonconvex quadratic problems, which has important applications, for example in high-dimensional statistical learning [16]. To proceed, we will assume the following.

  B-(a) Each function $g_i(z)$ is a quadratic function of the form $g_i(z) = 1/2z^T A_i z + \langle b, z \rangle$, where $A_i$ is a symmetric matrix but not necessarily positive semidefinite;

  B-(b) The feasible set $Z$ is a closed compact polyhedral set;

  B-(c) The nonsmooth function $p(z) = \mu\|z\|_1$, for some $\mu \geq 0$.

Our linear convergence result is based upon certain error bound condition around the stationary solutions set, which has been shown in [18] for smooth quadratic problems and has been extended to including $\ell_1$ penalty in [25, Theorem 4]. Due to space limitation the statement of the condition will be given in the supplemental material, along with the proof of the following result.

**Theorem 2.2.** *Suppose that Assumptions A, B are satisfied. Then the sequence $\{\mathbb{E}[Q^{r+1}]\}_{r=1}^\infty$ converges Q-linearly [4] to some $Q^* = f(z^*)$, where $z^*$ is a stationary solution for problem* (1.1). *That is, there exists a finite $\bar{r} > 0$, $\rho \in (0, 1)$ such that for all $r \geq \bar{r}$, $\mathbb{E}[Q^{r+1} - Q^*] \leq \rho\mathbb{E}[Q^r - Q^*]$.*

Linear convergence of this type for problems satisfying Assumption B has been shown for (deterministic) proximal gradient based methods [25, Theorem 2, 3]. To the best of our knowledge, this is the first result that shows the same linear convergence for a stochastic and distributed algorithm.

## 3   The NESTT-E Algorithm

**Algorithm Description.** In this section, we present a variant of NESTT-G, which is named NESTT with Exact minimization (NESTT-E). Our motivation is the following. First, in NESTT-G every agent should update its local variable at every iteration [cf. (2.4) or (2.6)]. In practice this may not be possible, for example at any given time a few agents can be in the *sleeping mode* so they cannot perform (2.6). Second, in the distributed setting it has been generally observed (e.g., see [8, Section V]) that performing exact minimization (whenever possible) instead of taking the gradient steps for local problems can significantly speed up the algorithm. The NESTT-E algorithm to be presented in this section is designed to address these issues. To proceed, let us define a new function as follows:

$$U(x, z; \lambda) := \sum_{i=1}^N U_i(x_i, z; \lambda_i) := \sum_{i=1}^N \left( \frac{1}{N} g_i(x_i) + \langle \lambda_i, x_i - z \rangle + \frac{\alpha_i\eta_i}{2}\|x_i - z\|^2 \right).$$

Note that if $\alpha_i = 1$ for all $i$, then the $L(x, z; \lambda) = U(x, z; \lambda) + p(z) + h(z)$. The algorithm details are presented in Algorithm 2.

**Convergence Analysis.** We begin analyzing NESTT-E. The proof technique is quite different from that for NESTT-G, and it is based upon using the expected value of the *Augmented Lagrangian* function as the potential function; see [11, 12, 13]. For the ease of description we define the following quantities:

$$w := (x, z, \lambda), \quad \beta := \frac{1}{\sum_{i=1}^{N} \eta_i}, \quad c_i := \frac{L_i^2}{\alpha_i \eta_i N^2} - \frac{\gamma_i}{2} + \frac{1 - \alpha_i}{\alpha_i} \frac{L_i}{N}, \quad \alpha := \{\alpha_i\}_{i=1}^{N}.$$

To measure the optimality of NESTT-E, define the *prox-gradient* of $L(x, z; \lambda)$ as:

$$\tilde{\nabla} L(w) = \left[ (z - \text{prox}_h[z - \nabla_z(L(w) - h(z))]); \nabla_{x_1} L(w); \cdots ; \nabla_{x_N} L(w) \right] \in \mathbb{R}^{(N+1)d}. \quad (3.19)$$

We define the optimality gap by adding to $\|\tilde{\nabla} L(w)\|^2$ the size of the constraint violation [13]:

$$H(w^r) := \|\tilde{\nabla} L(w^r)\|^2 + \sum_{i=1}^{N} \frac{L_i^2}{N^2} \|x_i^r - z^r\|^2.$$

It can be verified that $H(w^r) \to 0$ implies that $w^r$ reaches a stationary solution for problem (1.2). We have the following theorem regarding the convergence properties of NESTT-E.

**Theorem 3.1.** *Suppose Assumption A holds, and that $(\eta_i, \alpha_i)$ are chosen such that $c_i < 0$. Then for some constant $\underline{f}$, we have*

$$\mathbb{E}[L(w^r)] \geq \mathbb{E}[L(w^{r+1})] \geq \underline{f} > -\infty, \quad \forall r \geq 0.$$

*Further, almost surely every limit point of $\{w^r\}$ is a stationary solution of problem* (1.2)*. Finally, for some function of $\alpha$ denoted as $C(\alpha) = \sigma_1(\alpha)/\sigma_2(\alpha)$, we have the following:*

$$\mathbb{E}[H(w^m)] \leq \frac{C(\alpha)\mathbb{E}[L(w^1) - L(w^{R+1})]}{R}, \quad (3.20)$$

*where $\sigma_1 := \max(\hat{\sigma}_1(\alpha), \tilde{\sigma}_1)$ and $\sigma_2 := \max(\hat{\sigma}_2(\alpha), \tilde{\sigma}_2)$, and these constants are given by*

$$\hat{\sigma}_1(\alpha) = \max_i \left\{ 4 \left( \frac{L_i^2}{N^2} + \eta_i^2 + \left( \frac{1}{\alpha_i} - 1 \right)^2 \frac{L_i^2}{N^2} \right) + 3 \left( \frac{L_i^4}{\alpha_i \eta_i^2 N^4} + \frac{L_i^2}{N^2} \right) \right\},$$

$$\tilde{\sigma}_1 = \sum_{i=1}^{N} 4\eta_i^2 + (2 + \sum_{i=1}^{N} \eta_i + L_0)^2 + 3 \sum_{i=1}^{N} \frac{L_i^2}{N^2},$$

$$\hat{\sigma}_2(\alpha) = \max_i \left\{ p_i \left( \frac{\gamma_i}{2} - \frac{L_i^2}{N^2 \alpha_i \eta_i} - \frac{1 - \alpha_i}{\alpha_i} \frac{L_i}{N} \right) \right\}, \quad \tilde{\sigma}_2 = \frac{\sum_{i=1}^{N} \eta_i - L_0}{2}.$$

We remark that the above result shows the sublinear convergence of NESTT-E to the set of stationary solutions. Note that $\gamma_i = \eta_i - L_i/N$, to satisfy $c_i < 0$, a simple derivation yields

$$\eta_i > \frac{L_i \left( (2 - \alpha_i) + \sqrt{(\alpha_i - 2)^2 + 8\alpha_i} \right)}{2N\alpha_i}.$$

Further, the above result characterizes the dependency of the rates on various parameters of the algorithm. For example, to see the effect of $\alpha$ on the convergence rate, let us set $p_i = \frac{L_i}{\sum_{i=1}^{N} L_i}$, and $\eta_i = 3L_i/N$, and assume $L_0 = 0$, then consider two different choices of $\alpha$: $\hat{\alpha}_i = 1, \forall i$ and $\tilde{\alpha}_i = 4, \forall i$. One can easily check that applying these different choices leads to following results:

$$C(\hat{\alpha}) = 49 \sum_{i=1}^{N} L_i/N, \qquad C(\tilde{\alpha}) = 28 \sum_{i=1}^{N} L_i/N.$$

The key observation is that increasing $\alpha_i$'s reduces the constant in front of the rate. Hence, we expect that in practice larger $\alpha_i$'s will yield faster convergence.

## 4 Connections and Comparisons with Existing Works

In this section we compare NESTT-G/E with a few existing algorithms in the literature. First, we present a somewhat surprising observation, that NESTT-G takes the same form as some well-known algorithms for *convex* finite-sum problems. To formally state such relation, we show in the following result that NESTT-G in fact admits a compact *primal-only* characterization.

Table 1: Comparison of # of gradient evaluations for NESTT-G and GD in the worst case

| | NESTT-G | GD |
|---|---|---|
| **# of Gradient Evaluations** | $\mathcal{O}\left((\sum_{i=1}^{N}\sqrt{L_i/N})^2/\epsilon\right)$ | $\mathcal{O}\left(\sum_{i=1}^{N}L_i/\epsilon\right)$ |
| **Case I**: $L_i = 1,\ \forall i$ | $\mathcal{O}(N/\epsilon)$ | $\mathcal{O}(N/\epsilon)$ |
| **Case II** : $\mathcal{O}(\sqrt{N})$ terms with $L_i = N$ the rest with $L_i = 1$ | $\mathcal{O}(N/\epsilon)$ | $\mathcal{O}(N^{3/2}/\epsilon)$ |
| **Case III** : $\mathcal{O}(1)$ terms with $L_i = N^2$ the rest with $L_i = 1$ | $\mathcal{O}(N/\epsilon)$ | $\mathcal{O}(N^2/\epsilon)$ |

**Proposition 4.1.** *The NESTT-G can be written into the following compact form:*

$$z^{r+1} = \arg\min_{z}\ h(z) + g_0(z) + \frac{1}{2\beta}\|z - u^{r+1}\|^2 \tag{4.21a}$$

$$\text{with}\quad u^{r+1} := z^r - \beta\Big(\frac{1}{N\alpha_{i_r}}(\nabla g_{i_r}(z^r) - \nabla g_{i_r}(y_{i_r}^{r-1})) + \frac{1}{N}\sum_{i=1}^{N}\nabla g_i(y_i^{r-1})\Big). \tag{4.21b}$$

Based on this observation, the following comments are in order.

(1) Suppose $h \equiv 0$, $g_0 \equiv 0$ and $\alpha_i = 1$, $p_i = 1/N$ for all $i$. Then (4.21) takes the same form as the SAG presented in [22]. Further, when the component functions $g_i$'s are picked *cyclically* in a Gauss-Seidel manner, the iteration (4.21) takes the same form as the IAG algorithm [5].

(2) Suppose $h \neq 0$ and $g_0 \neq 0$, and $\alpha_i = p_i = 1/N$ for all $i$. Then (4.21) is the same as the SAGA algorithm [9], which is design for optimizing convex nonsmooth finite sum problems.

Note that SAG/SAGA/IAG are all designed for convex problems. Through the lens of primal-dual splitting, our work shows that they can be generalized to nonconvex nonsmooth problems as well.

Secondly, NESTT-E is related to the proximal version of the nonconvex ADMM [13, Algorithm 2]. However, the introduction of $\alpha_i$'s is new, which can significantly improve the practical performance but complicates the analysis. Further, there has been no counterpart of the sublinear and linear convergence rate analysis for the stochastic version of [13, Algorithm 2].

Thirdly, we note that a recent paper [21] has shown that SAGA works for smooth and unconstrained nonconvex problem. Suppose that $h \equiv 0$, $g_0 \neq 0$, $L_i = L_j$, $\forall\, i, j$ and $\alpha_i = p_i = 1/N$, the authors show that SAGA achieves $\epsilon$-stationarity using $\mathcal{O}(N^{2/3}(\sum_{i=1}^{N}L_i/N)/\epsilon)$ gradient evaluations. Compared with GD, which achieves $\epsilon$-stationarity using $\mathcal{O}(\sum_{i=1}^{N}L_i/\epsilon)$ gradient evaluations in the worse case (in the sense that $\sum_{i=1}^{N}L_i/N = L$), the rate in [21] is $\mathcal{O}(N^{1/3})$ times better. However, the algorithm in [21] is different from NESTT-G in two aspects: 1) it does not generalize to the nonsmooth constrained problem (1.1); 2) it samples two component functions at each iteration, while NESTT-G only samples once. Further, the analysis and the scaling are derived for the case of uniform $L_i$'s, therefore it is not clear how the algorithm and the rates can be adapted for the non-uniform case. On the other hand, our NESTT works for the general nonsmooth constrained setting. The non-uniform sampling used in NESTT-G is well-suited for problems with non-uniform $L_i$'s, and our scaling can be up to $N$ times better than GD (or its proximal version) in the worst case. Note that problems with non-uniform $L_i$'s for the component functions are common in applications such as sparse optimization and signal processing. For example in LASSO problem the data matrix is often normalized by feature (or "column-normalized" [19]), therefore the $\ell_2$ norm of each row of the data matrix (which corresponds to the Lipschitz constant for each component function) can be dramatically different.

In Table 1 we list the comparison of the number of gradient evaluations for NESTT-G and GD, in the worst case (in the sense that $\sum_{i=1}^{N}L_i/N = L$). For simplicity, we omitted an additive constant of $\mathcal{O}(N)$ for computing the initial gradients.

## 5 Numerical Results

In this section we evaluate the performance of NESTT. Consider the high dimensional regression problem with noisy observation [16], where $M$ observations are generated by $y = X\nu + \epsilon$. Here $y \in \mathbb{R}^M$ is the observed data sample; $X \in \mathbb{R}^{M \times P}$ is the covariate matrix; $\nu \in \mathbb{R}^P$ is the ground truth, and $\epsilon \in \mathbb{R}^M$ is the noise. Suppose that the covariate matrix is not perfectly known, i.e., we observe $A = X + W$ where $W \in \mathbb{R}^{M \times P}$ is the noise matrix with known covariance matrix $\Sigma_W$. Let us define $\hat{\Gamma} := 1/M(A^\top A) - \Sigma_W$, and $\hat{\gamma} := 1/M(A^\top y)$. To estimate the ground truth $\nu$, let

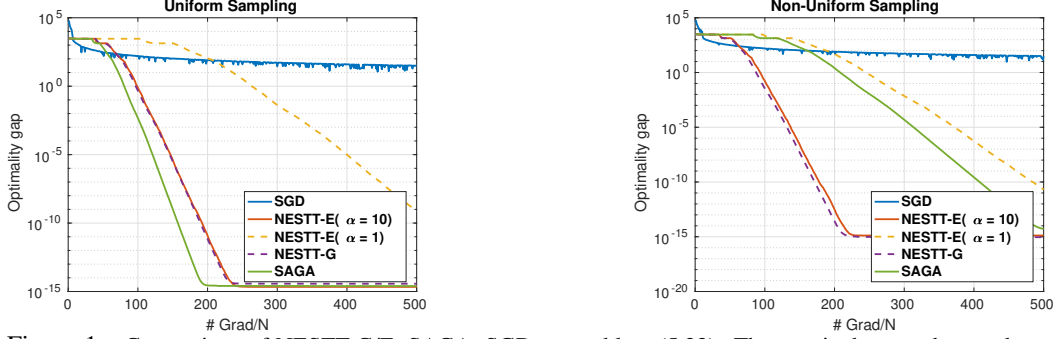

Figure 1: Comparison of NESTT-G/E, SAGA, SGD on problem (5.22). The $x$-axis denotes the number of passes of the dataset. Left: Uniform Sampling $p_i = 1/N$; Right: Non-uniform Sampling ($p_i = \frac{\sqrt{L_i/N}}{\sum_{i=1}^N \sqrt{L_i/N}}$).

Table 2: Optimality gap $\|\tilde{\nabla}_{1/\beta} f(z^r)\|^2$ for different algorithms, with 100 passes of the datasets.

| | SGD | | NESTT-E ($\alpha = 10$) | | NESTT-G | | SAGA | |
|---|---|---|---|---|---|---|---|---|
| N | Uniform | Non-Uni | Uniform | Non-Uni | Uniform | Non-Uni | Uniform | Non-Uni |
| 10 | 3.4054 | 0.2265 | 2.6E-16 | 6.16E-19 | 2.3E-21 | 6.1E-24 | 2.7E-17 | 2.8022 |
| 20 | 0.6370 | 6.9087 | 2.4E-9 | 5.9E-9 | 1.2E-10 | 2.9E-11 | 7.7E-7 | 11.3435 |
| 30 | 0.2260 | 0.1639 | 3.2E-6 | 2.7E-6 | 4.5E-7 | 1.4E-7 | 2.5E-5 | 0.1253 |
| 40 | 0.0574 | 0.3193 | 5.8E-4 | 8.1E-5 | 1.8E-5 | 3.1E-5 | 4.1E-5 | 0.7385 |
| 50 | 0.0154 | 0.0409 | 8.3E.-4 | 7.1E-4 | 1.2E-4 | 2.7E-4 | 2.5E-4 | 3.3187 |

us consider the following (nonconvex) optimization problem posed in [16, problem (2.4)] (where $R > 0$ controls sparsity):

$$\min_z \ z^\top \hat{\Gamma} z - \hat{\gamma} z \quad \text{s.t.} \quad \|z\|_1 \leq R. \tag{5.22}$$

Due to the existence of noise, $\hat{\Gamma}$ is not positive semidefinite hence the problem is not convex. Note that this problem satisfies Assumption A– B, then by Theorem 2.2 NESTT-G converges Q-linearly.

To test the performance of the proposed algorithm, we generate the problem following similar setups as [16]. Let $X = (X_1; \cdots, X_N) \in \mathbb{R}^{M \times P}$ with $\sum_i N_i = M$ and each $X_i \in \mathbb{R}^{N_i \times P}$ corresponds to $N_i$ data points, and it is generated from i.i.d Gaussian. Here $N_i$ represents the size of each mini-batch of samples. Generate the observations $y_i = X_i \times \nu^* + \epsilon_i \in \mathbb{R}^{N_i}$, where $\nu^*$ is a $K$-sparse vector to be estimated, and $\epsilon_i \in \mathbb{R}^{N_i}$ is the random noise. Let $W = [W_1; \cdots; W_N]$, with $W_i \in \mathbb{R}^{N_i \times P}$ generated with i.i.d Gaussian. Therefore we have $z^\top \hat{\Gamma} z = \frac{1}{N} \sum_{i=1}^N \frac{N}{M} z^\top \left( X_i^\top X_i - W_i^\top W_i \right) z$. We set $M = 100,000$, $P = 5000$, $N = 50$, $K = 22 \approx \sqrt{P}$, and $R = \|\nu^*\|_1$. We implement NESTT-G/E, the SGD, and the nonconvex SAGA proposed in [21] with stepsize $\beta = \frac{1}{3L_{\max} N^{2/3}}$ (with $L_{\max} := \max_i L_i$). Note that the SAGA proposed in [21] *only* works for the unconstrained problems with uniform $L_i$, therefore when applied to (5.22) it is *not* guaranteed to converge. Here we only include it for comparison purposes.

In Fig. 1 we compare different algorithms in terms of the gap $\|\tilde{\nabla}_{1/\beta} f(z^r)\|^2$. In the left figure we consider the problem with $N_i = N_j$ for all $i, j$, and we show performance of the proposed algorithms with uniform sampling (i.e., the probability of picking $i$th block is $p_i = 1/N$). On the right one we consider problems in which approximately half of the component functions have twice the size of $L_i$'s as the rest, and consider the non-uniform sampling ($p_i = \sqrt{L_i/N}/\sum_{i=1}^N \sqrt{L_i/N}$). Clearly in both cases the proposed algorithms perform quite well. Furthermore, it is clear that the NESTT-E performs well with large $\alpha := \{\alpha_i\}_{i=1}^N$, which confirms our theoretical rate analysis. Also it is worth mentioning that when the $N_i$'s are non-uniform, the proposed algorithms [NESTT-G and NESTT-E (with $\alpha = 10$)] significantly outperform SAGA and SGD. In Table 2 we further compare different algorithms when changing the number of component functions (i.e., the number of mini-batches $N$) while the rest of the setup is as above. We run each algorithm with 100 passes over the dataset. Similarly as before, our algorithms perform well, while SAGA seems to be sensitive to the uniformity of the size of the mini-batch [note that there is no convergence guarantee for SAGA applied to the nonconvex constrained problem (5.22)].

## Footnotes

[4]A sequence $\{x^r\}$ is said to converge Q-linearly to some $\bar{x}$ if $\limsup_r \|x^{r+1} - \bar{x}\|/\|x^r - \bar{x}\| \leq \rho$, where $\rho \in (0, 1)$ is some constant; cf [25] and references therein.

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
