[Supplementary Material]

# Appendix

## 5.1 Some Key Properties of NESTT-G

To facilitate the following derivation, in this section we collect some key properties of NESTT-G.

First, from the optimality condition of the $x$ update we have

$$x_{i_r}^{r+1} = z^r - \frac{1}{\alpha_{i_r}\eta_{i_r}}\left(\lambda_{i_r}^r + \frac{1}{N}\nabla g_{i_r}(z^r)\right), \tag{5.23a}$$

$$x_j^{r+1} \stackrel{(2.6)}{=} z^r \stackrel{(2.8b)}{=} z^r - \frac{1}{\alpha_j\eta_j}(\lambda_j^r + \frac{1}{N}\nabla g_j(z^{r(j)})), \ \forall \ j \neq i_r. \tag{5.23b}$$

Then using the update scheme of the $\lambda$ we can further obtain

$$\lambda_{i_r}^{r+1} = -\frac{1}{N}\nabla g_{i_r}(z^r), \tag{5.24a}$$

$$\lambda_j^{r+1} = -\frac{1}{N}\nabla g_j(z^{r(j)}), \ \forall \ j \neq i_r. \tag{5.24b}$$

Therefore, using the definition of $y_i^r$ we have the following compact forms

$$\lambda_i^{r+1} = -\frac{1}{N}\nabla g_i(y_i^r), \ i = 1, \cdots, N. \tag{5.25}$$

$$x_i^{r+1} = z^r - \frac{1}{\alpha_i\eta_i}\left(\lambda_i^r + \frac{1}{N}\nabla g_i(y_i^r)\right), \ i = 1, \cdots, N. \tag{5.26}$$

Second, let us look at the optimality condition for the $z$ update. The $z$-update (2.7) is given by

$$z^{r+1} = \arg\min_z \ L(\{x_i^{r+1}\}, z; \lambda^r)$$

$$= \arg\min_z \ \sum_{i=1}^N \left(\langle\lambda_i^r, x_i^{r+1} - z\rangle + \frac{\eta_i}{2}\|x_i^{r+1} - z\|^2\right) + g_0(z) + h(z). \tag{5.27}$$

Note that this problem is strongly convex because we have assumed that $\sum_{i=1}\eta_i > 3L_0$; cf. Assumption [A-(c)].

Let us define

$$u^{r+1} := \frac{\sum_{i=1}^N \eta_i x_i^{r+1} + \sum_{i=1}^N \lambda_i^r}{\sum_{i=1}^N \eta_i}$$

$$= \frac{\sum_{i=1}^N \eta_i z^r - \eta_{i_r}(z^r - x_{i_r}^{r+1})}{\sum_{i=1}^N \eta_i} + \frac{\sum_{i=1}^N \lambda_i^r}{\sum_{i=1}^N \eta_i}$$

$$\stackrel{(5.23a)}{=} \frac{\sum_{i=1}^N \eta_i z^r - \frac{\eta_{i_r}}{\alpha_{i_r}\eta_{i_r}}(\lambda_{i_r}^r + 1/N\nabla g_{i_r}(z^r))}{\sum_{i=1}^N \eta_i} + \frac{\sum_{i=1}^N \lambda_i^r}{\sum_{i=1}^N \eta_i}$$

$$\stackrel{(5.25)}{=} z^r - \frac{\frac{1}{\alpha_{i_r}}(-\nabla g_{i_r}(y_{i_r}^{r-1}) + \nabla g_{i_r}(z^r))}{N\sum_{i=1}^N \eta_i} - \frac{\sum_{i=1}^N \nabla g_i(y_i^{r-1})}{N\sum_{i=1}^N \eta_i}$$

$$\stackrel{(i)}{=} z^r - \frac{\beta}{N\alpha_{i_r}}(-\nabla g_{i_r}(y_{i_r}^{r-1}) + \nabla g_{i_r}(z^r)) - \frac{\beta\sum_{i=1}^N \nabla g_i(y_i^{r-1})}{N} \tag{5.28}$$

$$\stackrel{(ii)}{:=} z^r - \beta v_{i_r}^{r+1} \tag{5.29}$$

where in (i) we have defined $\beta := 1/\sum_{i=1}^N \eta_i$; in (ii) we have defined

$$v_{i_r}^{r+1} := \frac{1}{N}\sum_{i=1}^N \nabla g_i(y_i^{r-1}) + \frac{1}{\alpha_{i_r}}\left(-\frac{1}{N}\nabla g_{i_r}(y_{i_r}^{r-1}) + \frac{1}{N}\nabla g_{i_r}(z^r)\right). \tag{5.30}$$

Clearly if we pick $\alpha_i = p_i$ for all $i$, then we have

$$\mathbb{E}_{i_r}[u^{r+1} \mid \mathcal{F}^r] = z^r - \frac{\beta}{N} \sum_{i=1}^{N} \nabla g_i(z^r). \tag{5.31}$$

Using the definition of $u^{r+1}$, it is easy to check that

$$\begin{aligned}
z^{r+1} &= \arg\min_z \frac{1}{2\beta} \|z - u^{r+1}\|^2 + h(z) + g_0(z) \\
&= \mathrm{prox}_h^{1/\beta}[u^{r+1} - \beta \nabla g_0(z^{r+1})]. 
\end{aligned} \tag{5.32}$$

The optimality condition for the $z$ subproblem is given by:

$$z^{r+1} - u^{r+1} + \beta \nabla g_0(z^{r+1}) + \beta \xi^{r+1} = 0 \tag{5.33}$$

where, $\xi^{r+1} \in \partial h(z^{r+1})$ is a subgradient of $h(z^{r+1})$. Using the definition of $v_{i_r}$ in (5.30), we obtain

$$z^{r+1} = z^r - \beta(v_{i_r}^{r+1} + \nabla g_0(z^{r+1}) + \xi^{r+1}). \tag{5.34}$$

Third, if $\alpha_i = p_i$, then we have:

$$\begin{aligned}
\mathbb{E}_{i_r} &\left[ \left\| -\frac{\lambda_{i_r}^r + 1/N \nabla g_{i_r}(z^r)}{\alpha_{i_r}} + \frac{1}{N} \sum_{i=1}^{N} \nabla g_i(z^r) - \sum_{i=1}^{N} \frac{1}{N} \nabla g_i(y_i^{r-1}) \right\|^2 \right] \\
&\stackrel{(a)}{=} \mathrm{Var}\left[ -\frac{\lambda_{i_r}^r + 1/N \nabla g_{i_r}(z^r)}{\alpha_{i_r}} \right] \\
&\stackrel{(b)}{\leq} \sum_{i=1}^{N} \frac{1}{\alpha_i} \left\| \frac{1}{N} \nabla g_i(z^r) - \frac{1}{N} \nabla g_i(y_i^{r-1}) \right\|^2,
\end{aligned} \tag{5.35}$$

where $(a)$ is true because whenever $\alpha_i = p_i$ for all $i$, then

$$\mathbb{E}_{i_r}\left[ -\frac{\lambda_{i_r}^r + 1/N \nabla g_{i_r}(z^r)}{\alpha_{i_r}} \right] = \frac{1}{N} \sum_{i=1}^{N} \nabla g_i(z^r) - \sum_{i=1}^{N} \frac{1}{N} \nabla g_i(y_i^{r-1});$$

The inequality in $(b)$ is true because for a random variable $x$ we have $\mathrm{Var}(x) \leq \mathbb{E}[x^2]$.

## 5.2 Proof of Lemma 2.1

**Step 1).** Using the definition of potential function $Q^r$, we have:

$$\mathbb{E}[Q^r - Q^{r-1} \mid \mathcal{F}^{r-1}]$$
$$= \mathbb{E}\left[ \sum_{i=1}^{N} \frac{1}{N} (g_i(z^r) - g_i(z^{r-1})) + g_0(z^r) - g_0(z^{r-1}) + h(z^r) - h(z^{r-1}) \mid \mathcal{F}^{r-1} \right]$$
$$+ \mathbb{E}\left[ \sum_{i=1}^{N} \frac{3p_i}{\alpha_i^2 \eta_i} \left\| \frac{1}{N} \nabla g_i(z^r) - \frac{1}{N} \nabla g_i(y_i^{r-1}) \right\|^2 - \frac{3p_i}{\alpha_i^2 \eta_i} \left\| \frac{1}{N} \nabla g_i(z^{r-1}) - \frac{1}{N} \nabla g_i(y_i^{r-2}) \right\|^2 \mid \mathcal{F}^{r-1} \right]. \tag{5.36}$$

**Step 2).** The first term in (5.36) can be bounded as follows (omitting the subscript $\mathcal{F}^r$).

$$
\mathbb{E}\left[\sum_{i=1}^{N} \frac{1}{N}\left(g_i(z^r) - g_i(z^{r-1})\right) + g_0(z^r) - g_0(z^{r-1}) + h(z^r) - h(z^{r-1}) \mid \mathcal{F}^{r-1}\right]
$$

$$
\overset{(i)}{\leq} \mathbb{E}\left[\frac{1}{N}\sum_{i=1}^{N}\langle \nabla g_i(z^{r-1}), z^r - z^{r-1}\rangle + \langle \nabla g_0(z^{r-1}), z^r - z^{r-1}\rangle\right.
$$

$$
\left. + \langle \xi^r, z^r - z^{r-1}\rangle + \frac{\sum_{i=1}^{N} L_i/N + L_0}{2}\|z^r - z^{r-1}\|^2 \mid \mathcal{F}^{r-1}\right]
$$

$$
\overset{(ii)}{=} \mathbb{E}\left[\left\langle \frac{1}{N}\sum_{i=1}^{N}\nabla g_i(z^{r-1}) + \xi^r + \nabla g_0(z^r) + \frac{1}{\beta}(z^r - z^{r-1}), z^r - z^{r-1}\right\rangle \mid \mathcal{F}^{r-1}\right]
$$

$$
- \left(\frac{1}{\beta} - \frac{\sum_{i=1}^{N} L_i/N + 3L_0}{2}\right)\mathbb{E}_{z^r}\|z^r - z^{r-1}\|^2
$$

$$
\overset{(5.34)}{=} \mathbb{E}\left[\left\langle \frac{1}{N}\sum_{i=1}^{N}\nabla g_i(z^{r-1}) - v_{i(r-1)}^r, z^r - z^{r-1}\right\rangle \mid \mathcal{F}^{r-1}\right]
$$

$$
- \left(\frac{1}{\beta} - \frac{\sum_{i=1}^{N} L_i/N + 3L_0}{2}\right)\mathbb{E}_{z^r}\|z^r - z^{r-1}\|^2
$$

$$
\overset{(iii)}{\leq} \frac{1}{2\ell_1}\mathbb{E}\left[\left\|1/N\sum_{i=1}^{N}\nabla g_i(z^{r-1}) - v_{i(r-1)}^r\right\|^2 \mid \mathcal{F}^{r-1}\right] + \frac{\ell_1}{2}\mathbb{E}_{z^r}\|z^r - z^{r-1}\|^2
$$

$$
- \left(\frac{1}{\beta} - \frac{\sum_{i=1}^{N} L_i/N + 3L_0}{2}\right)\mathbb{E}_{z^r}\|z^r - z^{r-1}\|^2 \tag{5.37}
$$

where in (i) we have used the Lipschitz continuity of the gradients of $g_i$'s as well as the convexity of $h$; in (ii) we have used the fact that

$$
\langle \nabla g_0(z^{r-1}), z^r - z^{r-1}\rangle \leq \langle \nabla g_0(z^r), z^r - z^{r-1}\rangle + L_0\|z^r - z^{r-1}\|^2; \tag{5.38}
$$

in (iii) we have applied the Young's inequality for some $\ell_1 > 0$.

Choosing $\ell_1 = \frac{1}{2\beta}$, we have:

$$
\frac{1}{2\ell_1}\mathbb{E}\left\|\frac{1}{N}\sum_{i=1}^{N}\nabla g_i(z^{r-1}) - v_{i(r-1)}^r\right\|^2
$$

$$
\overset{(5.30)}{=} \beta\mathbb{E}\left[\left\|\frac{1}{N}\sum_{i=1}^{N}\nabla g_i(z^{r-1}) - \frac{\lambda_{i(r-1)}^{r-1} + 1/N\nabla g_{i(r-1)}(z^{r-1})}{\alpha_{i(r-1)}} - \sum_{i=1}^{N}\frac{1}{N}\nabla g_i(y_i^{r-2})\right\|^2\right]
$$

$$
\overset{(5.35)}{\leq} \beta\sum_{i=1}^{N}\frac{1}{\alpha_i}\left\|\frac{1}{N}\nabla g_i(z^{r-1}) - \frac{1}{N}\nabla g_i(y_i^{r-2})\right\|^2.
$$

Overall we have the following bound for the first term in (5.36):

$$
\mathbb{E}\left[\sum_{i=1}^{N} \frac{1}{N}\left(g_i(z^r) - g_i(z^{r-1})\right) + g_0(z^r) - g_0(z^{r-1}) + h(z^r) - h(z^{r-1}) \mid \mathcal{F}^{r-1}\right] \tag{5.39}
$$

$$
\leq \sum_{i=1}^{N}\frac{\beta}{\alpha_i}\left\|\frac{1}{N}\nabla g_i(z^{r-1}) - \frac{1}{N}\nabla g_i(y_i^{r-2})\right\|^2 - \left(\frac{3}{4\beta} - \frac{\sum_{i=1}^{N} L_i/N + 3L_0}{2}\right)\mathbb{E}_{z^r}\|z^r - z^{r-1}\|^2.
$$

**Step 3).** We bound the second term in (5.36) in the following way:

$$\mathbb{E}\left[\|\nabla g_i(z^r) - \nabla g_i(y_i^{r-1})\|^2 \mid \mathcal{F}^{r-1}\right]$$

$$= \mathbb{E}\left[\|\nabla g_i(z^r) - \nabla g_i(y_i^{r-1}) + \nabla g_i(z^{r-1}) - \nabla g_i(z^{r-1})\|^2 \mid \mathcal{F}^{r-1}\right]$$

$$\overset{(i)}{\leq} (1 + \xi_i)\mathbb{E}_{z^r}\|\nabla g_i(z^r) - \nabla g_i(z^{r-1})\|^2 + \left(1 + \frac{1}{\xi_i}\right)\mathbb{E}_{y_i^{r-1}}\|\nabla g_i(y_i^{r-1}) - \nabla g_i(z^{r-1})\|^2$$

$$\overset{(ii)}{=} (1 + \xi_i)\mathbb{E}_{z^r}\|\nabla g_i(z^r) - \nabla g_i(z^{r-1})\|^2 + (1 - p_i)\left(1 + \frac{1}{\xi_i}\right)\|\nabla g_i(y_i^{r-2}) - \nabla g_i(z^{r-1})\|^2$$

$$\tag{5.40}$$

where in (i) we have used the fact that the randomness of $z^{r-1}$ comes from $i_{r-2}$, so fixing $\mathcal{F}^{r-1}$, $z^{r-1}$ is deterministic; we have also applied the following inequality:

$$(a + b)^2 \leq (1 + \xi)a^2 + (1 + \frac{1}{\xi})b^2 \quad \forall\, \xi > 0.$$

The equality (ii) is true because the randomness of $y_i^{r-1}$ comes from $i_{r-1}$, and for each $i$ there is a probability $p_i$ such that $x_i^r$ is updated, so that $\nabla g_i(y_i^{r-1}) = \nabla g_i(z^{r-1})$, otherwise $x_i$ is not updated so that $\nabla g_i(y_i^{r-1}) = \nabla g_i(y_i^{r-2})$.

**Step 4).** Applying (5.40) and set $\alpha_i = p_i$, the second part of (5.36) can be bounded as

$$\mathbb{E}\left[\sum_{i=1}^N \frac{3p_i}{\alpha_i^2 \eta_i}\left\|\frac{1}{N}\nabla g_i(z^r) - \frac{1}{N}\nabla g_i(y_i^{r-1})\right\|^2 - \frac{3p_i}{\alpha_i^2 \eta_i}\left\|\frac{1}{N}\nabla g_i(z^{r-1}) - \frac{1}{N}\nabla g_i(y_i^{r-2})\right\|^2 \mid \mathcal{F}^{r-1}\right]$$

$$\leq \sum_{i=1}^N \frac{3L_i^2}{\alpha_i \eta_i N^2}(1 + \xi_i)\,\mathbb{E}_{z^r}\|z^r - z^{r-1}\|^2$$

$$+ \frac{3}{\alpha_i \eta_i}\left((1 - p_i)(1 + \frac{1}{\xi_i}) - 1\right)\left\|\frac{1}{N}\nabla g_i(y_i^{r-2}) - \frac{1}{N}\nabla g_i(z^{r-1})\right\|^2. \tag{5.41}$$

Combining (5.39) and (5.41) eventually we have

$$\mathbb{E}[Q^r - Q^{r-1} \mid \mathcal{F}^r]$$

$$\leq \sum_{i=1}^N \left\{\frac{\beta}{\alpha_i} + \frac{3}{\alpha_i \eta_i}\left((1 - p_i)(1 + \frac{1}{\xi_i}) - 1\right)\right\}\left\|\frac{1}{N}\nabla g_i(z^{r-1}) - \frac{1}{N}\nabla g_i(y_i^{r-2})\right\|^2$$

$$+ \left\{-\frac{3}{4\beta} + \frac{\sum_{i=1}^N L_i/N + 3L_0}{2} + \sum_{i=1}^N \frac{3L_i^2}{\alpha_i \eta_i N^2}(1 + \xi_i)\right\}\mathbb{E}_{z^r}\|z^r - z^{r-1}\|^2. \tag{5.42}$$

Let us define $\{\tilde{c}_i\}$ and $\hat{c}$ as following:

$$\tilde{c}_i = \frac{\beta}{\alpha_i} + \frac{3}{\alpha_i \eta_i}\left((1 - p_i)(1 + \frac{1}{\xi_i}) - 1\right)$$

$$\hat{c} = -\frac{3}{4\beta} + \frac{\sum_{i=1}^N L_i/N + 3L_0}{2} + \sum_{i=1}^N \frac{3L_i^2}{\alpha_i \eta_i N^2}(1 + \xi_i).$$

In order to prove the lemma it is enough to show that $\tilde{c}_i < -\frac{1}{2\eta_i}\ \forall\, i$, and $\hat{c} < -\sum_{i=1}^N \frac{\eta_i}{8}$. Let us pick

$$\alpha_i = p_i, \ \xi_i = \frac{2}{p_i}, \ p_i = \frac{\eta_i}{\sum_{i=1}^N \eta_i}. \tag{5.43}$$

Recall that $\beta = \frac{1}{\sum_{i=1}^N \eta_i}$. These values yield the following

$$\tilde{c}_i = \frac{1}{\eta_i} - \frac{3}{\eta_i}\left(\frac{p_i + 1}{2}\right) \leq \frac{1}{\eta_i} - \frac{3}{2\eta_i} = -\frac{1}{2\eta_i} < 0.$$

To show that $\hat{c} \le -\sum_{i=1}^{N} \frac{\eta_i}{8}$ let us assume that $\eta_i = d_i L_i$ for some $d_i > 0$. Note that by assumption we have

$$\sum_{i=1}^{N} \eta_i \ge 3L_0.$$

Therefore we have the following expression for $\hat{c}$:

$$\hat{c} \le -\sum_{i=1}^{N} \frac{1}{4} d_i L_i + \frac{L_i}{2N} + \frac{3L_i}{p_i d_i N^2}\left(1 + \frac{2}{p_i}\right)$$

$$< \sum_{i=1}^{N} \frac{L_i}{d_i}\left(-\frac{1}{4}d_i^2 + \frac{d_i}{2N} + \frac{9}{p_i^2 N^2}\right).$$

As a result, to have $\hat{c} < -\sum_{i=1}^{N} \frac{\eta_i}{8}$, we need

$$\frac{L_i}{d_i}\left(\frac{1}{4}d_i^2 - \frac{d_i}{2N} - \frac{9}{p_i^2 N^2}\right) \ge \frac{d_i L_i}{8}, \quad \forall\, i. \tag{5.44}$$

Or equivalently

$$\frac{1}{8}d_i^2 - \frac{d_i}{2N} - \frac{9}{p_i^2 N^2} \ge 0, \quad \forall\, i. \tag{5.45}$$

By finding the root of the above quadratic inequality, we need $d_i \ge \frac{9}{N p_i}$, which is equivalent to choosing the following parameters

$$\eta_i \ge \frac{9 L_i}{N p_i}. \tag{5.46}$$

The lemma is proved. **Q.E.D.**

## 5.3 Proof of Theorem 2.1

First, using the fact that $f(z)$ is lower bounded [cf. Assumption A-(a)], it is easy to verify that $\{Q^r\}$ is a bounded sequence. Denote its lower bound to be Q. From Lemma 2.1, it is clear that $\{Q^r - Q\}$ is a nonnegative supermartingale. Apply the Supermartigale Convergence Theorem [R1, Proposition 4.2] we conclude that $\{Q^r\}$ converges almost surely (a.s.), and that

$$\left\|\nabla g_i(z^{r-1}) - \nabla g_i(y_i^{r-2})\right\|^2 \to 0, \quad \mathbb{E}_{z^r}\|z^r - z^{r-1}\| \to 0, \quad \text{a.s.}, \quad \forall\, i. \tag{5.47}$$

The first inequality implies that $\|\lambda_{i_r}^r - \lambda_{i_r}^{r-1}\| \to 0$. Combining this with equation (2.5) yields $\|x_{i_r}^r - z^{r-1}\| \to 0$, which further implies that $\|z^r - z^{r-1}\| \to 0$. By utilizing (2.8b) – (2.8c), we can conclude that

$$\|x_i^r - x_i^{r-1}\| \to 0, \quad \|\lambda_i^r - \lambda_i^{r-1}\| \to 0, \quad \text{a.s.}, \quad \forall\, i. \tag{5.48}$$

That is, almost surely the successive differences of all the primal and dual variables go to zero. Then it is easy to show that every limit point of the sequence $(x^r, z^r, \lambda^r)$ converge to a stationary solution of problem (1.2) (for example, see the argument in [R2, Theorem 2.1]. Here we omit the full proof.

**Part 1).** We bound the gap in the following way (where the expectation is taking over the nature history of the algorithm):

$$\mathbb{E}\left[\|z^r - \text{prox}_h^{1/\beta}[z^r - \beta\nabla(g(z^r) + g_0(z^r))]\|^2\right]$$

$$\overset{(a)}{=} \mathbb{E}\left[\|z^r - z^{r+1} + \text{prox}_h^{1/\beta}[u^{r+1} - \beta\nabla g_0(z^{r+1})] - \text{prox}_h^{1/\beta}[z^r - \beta\nabla(g(z^r) + g_0(z^r))]\|^2\right]$$

$$\overset{(b)}{\leq} 3\mathbb{E}\|z^r - z^{r+1}\|^2 + 3\mathbb{E}\|u^{r+1} - z^r + \beta\nabla g(z^r)\|^2 + 3L_0^2\beta^2\|z^{r+1} - z^r\|^2$$

$$\overset{(c)}{\leq} \frac{10}{3}\mathbb{E}\|z^r - z^{r+1}\|^2 + 3\beta^2\mathbb{E}\left[\|\nabla g(z^r) - \frac{\lambda_{i_r}^r + 1/N\nabla g_{i_r}(z^r)}{\alpha_{i_r}} - \sum_{i=1}^N 1/N\nabla g_i(y_i^{r-1})\|^2\right]$$

$$\overset{(5.35)}{\leq} \frac{10}{3}\mathbb{E}\|z^r - z^{r+1}\|^2 + 3\beta^2\sum_{i=1}^N \frac{1}{\alpha_i}\mathbb{E}\left\|\frac{1}{N}\nabla g_i(z^r) - \frac{1}{N}\nabla g_i(y_i^{r-1})\right\|^2$$

$$\leq \frac{10}{3}\mathbb{E}\|z^r - z^{r+1}\|^2 + 3\sum_{i=1}^N \frac{\beta}{\eta_i}\mathbb{E}\left\|\frac{1}{N}\nabla g_i(z^r) - \frac{1}{N}\nabla g_i(y_i^{r-1})\right\|^2 \tag{5.49}$$

where $(a)$ is due to (5.32); $(b)$ is true due to the nonexpansivness of the prox operator, and the Cauchy-Swartz inequality; in $(c)$ we have used the definition of $u$ in (5.29) and the fact that $3L_0 \leq \sum_{i=1}^N \eta_i = \frac{1}{\beta}$ [cf. Assumption A-(c)]. In the last inequality we have applied (5.43), which implies that

$$\frac{\beta}{\alpha_i} = \frac{1}{p_i \sum_{j=1}^N \eta_j} = \frac{1}{\eta_i}. \tag{5.50}$$

Note that $\eta_i$'s has to satisfy (5.46). Let us follow (2.11) and choose

$$\eta_i = \frac{9L_i}{p_i N} = \frac{9\sum_{j=1}^N \eta_j}{N\eta_i}L_i.$$

We have

$$\eta_i = \sqrt{9L_i/N \sum_{j=1}^N \eta_j} = \sqrt{9L_i/N}\sqrt{\sum_{j=1}^N \eta_j} \tag{5.51}$$

Summing $i$ from 1 to $N$ we have

$$\sqrt{\sum_{i=1}^N \eta_i} = \sum_{i=1}^N \sqrt{9L_i/N} \tag{5.52}$$

Then we conclude that

$$\frac{1}{\beta} = \sum_{i=1}^N \eta_i = \left(\sum_{i=1}^N \sqrt{9L_i/N}\right)^2. \tag{5.53}$$

So plugging the expression of $\beta$ into (5.50) and (5.51), we conclude

$$\alpha_i = p_i = \frac{\sqrt{L_i/N}}{\sum_{i=1}^N \sqrt{L_i/N}}, \quad \eta_i = \sqrt{9L_i/N}\sum_{j=1}^N \sqrt{9L_j/N}. \tag{5.54}$$

After plugging in the above inequty into (2.13), we obtain:

$$\mathbb{E}[G^r] \overset{(5.49)}{\leq} \frac{10}{3\beta^2}\mathbb{E}\|z^r - z^{r+1}\|^2 + \sum_{i=1}^N \frac{3}{\beta\eta_i}\mathbb{E}\left\|\frac{1}{N}\nabla g_i(z^r) - \frac{1}{N}\nabla g_i(y_i^{r-1})\right\|^2 \tag{5.55}$$

$$\overset{(2.12)}{\leq} \frac{80}{3\beta}\mathbb{E}[Q^r - Q^{r+1}] = \frac{80}{3}\left(\sum_{i=1}^N \sqrt{L_i/N}\right)^2 \mathbb{E}[Q^r - Q^{r+1}]$$

If we sum both sides over $r = 1, \cdots, R$, we obtain:

$$\sum_{r=1}^{R} \mathbb{E}[G^r] \leq \frac{80}{3} \left( \sum_{i=1}^{N} \sqrt{L_i/N} \right)^2 \mathbb{E}[Q^1 - Q^{R+1}].$$

Using the definition of $z^m$, we have

$$\mathbb{E}[G^m] = \mathbb{E}_{\mathcal{F}^r} \left[ \mathbb{E}_m[G^m \mid \mathcal{F}^r] \right] = 1/R \sum_{r=1}^{R} \mathbb{E}_{\mathcal{F}^r}[G^r].$$

Therefore, we can finally conclude that:

$$\mathbb{E}[G^m] \leq \frac{80}{3} \left( \sum_{i=1}^{N} \sqrt{L_i/N} \right)^2 \frac{\mathbb{E}[Q^1 - Q^{R+1}]}{R} \tag{5.56}$$

which proves the first part.

**Part 2).** In order to prove the second part let us recycle inequality in (5.55) and write

$$\mathbb{E}\left[ G^r + \sum_{i=1}^{N} \frac{3}{\beta \eta_i} \left\| \frac{1}{N} \nabla g_i(z^r) - \frac{1}{N} \nabla g_i(y_i^{r-1}) \right\|^2 \right]$$

$$\leq \frac{10}{3\beta^2} \mathbb{E}\|z^{r+1} - z^r\|^2 + \sum_{i=1}^{N} \frac{6}{\beta \eta_i} \mathbb{E} \left\| \frac{1}{N} \nabla g_i(z^r) - \frac{1}{N} \nabla g_i(y_i^{r-1}) \right\|^2$$

$$\leq \frac{80}{3\beta} \mathbb{E}[Q^r - Q^{r+1}] = 48 \left( \sum_{i=1}^{N} \sqrt{L_i/N} \right)^2 \mathbb{E}[Q^r - Q^{r+1}].$$

Also note that

$$\mathbb{E}_{x^r} \left[ \left\| x_i^{r+1} - z^r \right\|^2 \mid \mathcal{F}^r \right] = \sum_{i=1}^{N} \frac{1}{\alpha_i \eta_i^2} \left\| \frac{1}{N} \nabla g_i(z^r) - \frac{1}{N} \nabla g_i(y_i^{r-1}) \right\|^2 \tag{5.57}$$

Combining the above two inequalities, we conclude

$$\mathbb{E}_{\mathcal{F}^r}[G^r] + \mathbb{E}_{\mathcal{F}^r} \left[ \sum_{i=1}^{N} 3\eta_i^2 \left\| x_i^{r+1} - z^r \right\|^2 \right]$$

$$= \mathbb{E}_{\mathcal{F}^r}[G^r] + \mathbb{E}_{\mathcal{F}^r} \left[ \sum_{i=1}^{N} \frac{3\eta_i \alpha_i}{\beta} \left\| x_i^{r+1} - z^r \right\|^2 \right]$$

$$= \mathbb{E}\left[ G^r + \sum_{i=1}^{N} \frac{3}{\beta \eta_i} \left\| \frac{1}{N} \nabla g_i(z^r) - \frac{1}{N} \nabla g_i(y_i^{r-1}) \right\|^2 \right]$$

$$\leq \frac{80}{3} \left( \sum_{i=1}^{N} \sqrt{L_i/N} \right)^2 \mathbb{E}_{\mathcal{F}^r}[Q^r - Q^{r+1}] \tag{5.58}$$

where in the first equality we have used the relation $\frac{\alpha_i}{\beta} = \eta_i$ [cf. (5.50)]. Using a similar argument as in first part, we conclude that

$$\mathbb{E}[G^m] + \mathbb{E}\left[ \sum_{i=1}^{N} 3\eta_i^2 \left\| x_i^m - z^{m-1} \right\|^2 \right] \leq \frac{80}{3} \left( \sum_{i=1}^{N} \sqrt{L_i/N} \right)^2 \frac{\mathbb{E}[Q^1 - Q^{R+1}]}{R}. \tag{5.59}$$

This completes the proof. **Q.E.D.**

## 5.4 Proof of Theorem 2.2

We first need the following lemma, which characterizes certain error bound condition around the stationary solution set.

**Lemma 5.1.** *Suppose Assumptions A and B hold. Let $Z^*$ denotes the set of stationary solutions of problem* (1.1)*, and dist $(z, Z^*) := \min_{u \in Z^*} \|z - u\|$. Then we have the following*

1. (**Error Bound Condition**) *For any $\xi \geq \min_z f(z)$, exists a positive scalar $\tau$ such that the following error bound holds*

$$dist\,(z, Z^*) \leq \tau \|\tilde{\nabla}_{1/\beta} f(z)\| \tag{5.60}$$

*for all $z \in (Z \cap \operatorname{dom} h)$ and $z \in \{z : f(z) \leq \xi\}$.*

2. (**Separation of Isocost Surfaces**) *There exists a scalar $\delta > 0$ such that*

$$\|z - v\| \geq \delta \quad whenever \quad z \in Z^*, v \in Z^*, f(z) \neq f(v). \tag{5.61}$$

The first statement holds true largely due to [R3, Theorem 4], and the second statement holds true due to [R4, Lemma 3.1]; see detailed discussion after [R3, Assumption 2]. Here the only difference with the statement [R3, Theorem 4] is that the error bound condition (5.60) holds true *globally*. This is by the assumption that $Z$ is a compact set. Below we provide a brief argument.

From [R3, Theorem 4], we know that when Assumption B is satisfied, we have that for any $\xi \geq \min_z f(z)$, there exists scalars $\tau$ and $\epsilon$ such that the following error bound holds

$$\operatorname{dist}(z, Z^*) \leq \tau \|\tilde{\nabla}_{1/\beta} f(z)\|, \quad \text{whenever } \|\tilde{\nabla}_{1/\beta} f(z)\| \leq \epsilon, \ f(z) \leq \xi. \tag{5.62}$$

To argue that when $Z$ is compact, the above error bound is independent of $\epsilon$, we use the following two steps: (1) for all $z \in Z \cap \operatorname{dom}(h)$ such that $\|\tilde{\nabla}_{1/\beta} f(z)\| \leq \delta$, it is clear that the error bound (5.60) holds true; (2) for all $z \in Z \cap \operatorname{dom}(h)$ such that $\|\tilde{\nabla}_{1/\beta} f(z)\| \geq \delta$, the ratio $\frac{\operatorname{dist}(z, Z^*)}{\|\tilde{\nabla}_{1/\beta} f(z)\|}$ is a continuous function and well defined over the compact set $Z \cap \operatorname{dom}(h) \cap \left\{ z \mid \|\tilde{\nabla}_{1/\beta} f(z)\| \geq \delta \right\}$. Thus, the above ratio must be bounded from above by a constant $\tau'$ (independent of $b$, and no greater than $\max_{z, z' \in Z} \|z - z'\|/\delta$). Combining (1) and (2) yields the desired error bound over the set $Z \cap \operatorname{dom}(h)$. **Q.E.D.**

### Proof of Theorem 2.2

From Theorem 2.1 we know that $(x^r, z^r, \lambda^r)$ converges to the set of stationary solutions of problem (1.2). Let $(x^*, z^*, \lambda^*)$ be one of such stationary solution. Then by the definition of the $Q$ function and the fact that the successive differences of the gradients goes to zero (cf. (5.47)), we have

$$Q^* = f(z^*) = \sum_{i=1}^{N} 1/N g_i(z^*) + g_0(z^*) + p(z^*). \tag{5.63}$$

Then by Lemma 5.1 - (2) we know that $f(z^r) = \sum_{i=1}^{N} 1/N g_i(z^r) + g_0(z^r) + p(z^r)$ will finally settle at some isocost surface of $f$, i.e., there exists some *finite* $\bar{r} > 0$ such that for all $r > \bar{r}$ and $\bar{v} \in \mathbb{R}$ such that

$$f(\bar{z}^r) = \bar{v}, \quad \forall\, r \geq \bar{r} \tag{5.64}$$

where $\bar{z}^r = \arg \min_{z \in Z^*} \|z^r - z\|$. Therefore, combining the fact that $\|x^{r+1} - x^r\| \to 0$, $\|z^{r+1} - z^r\| \to 0$, $\|x_i^{r+1} - z^{r+1}\| \to 0$ and $\|\lambda^{r+1} - \lambda^r\| \to 0$ (cf. (5.87), (5.88)), it is easy to see that

$$L(\bar{z}^r, \bar{x}^r, \bar{\lambda}^r) = f(\bar{z}^r) = \bar{v}, \quad \forall\, r \geq \bar{r}, \tag{5.65}$$

where $\bar{x}^r, \bar{\lambda}^r$ are defined similarly as $\bar{z}^r$.

Now we prove that the expectation of $\Delta^{r+1} := Q^{r+1} - \bar{v}$ diminishes Q-linearly. All the expectation below is w.r.t. the natural history of the algorithm. The proof consists of the following steps:
**Step 1:** There exists $\sigma_1 > 0$ such that

$$\mathbb{E}[Q^r - Q^{r+1}] \geq \sigma_1 \left( \mathbb{E}\|z^{r+1} - z^r\|^2 + \sum_{i=1}^{N} \mathbb{E}\|1/N \nabla g_i(z^r) - 1/N \nabla g_i(y_i^{r-1})\|^2 \right);$$

**Step 2:** There exists $\tau > 0$ such that

$$\mathbb{E}\|z^r - \bar{z}^r\|^2 \leq \tau \|\mathbb{E}[\nabla_{1/\beta}\tilde{f}(z^r)]\|^2;$$

**Step 3:** There exists $\sigma_2 > 0$ such that

$$\|\mathbb{E}[\nabla_{1/\beta}\tilde{f}(z^r)]\|^2 \leq \sigma_2 \left( \mathbb{E}\|z^{r+1} - z^r\|^2 + \sum_{i=1}^{N} \mathbb{E}\|1/N\nabla g_i(z^r) - 1/N\nabla g_i(y_i^{r-1})\|^2 \right);$$

**Step 4:** There exists $\sigma_3 > 0$ such that the following relation holds true for all $r \geq \bar{r}$

$$\mathbb{E}[Q^{r+1} - \bar{v}] \leq \sigma_3 \left( \mathbb{E}\|z^r - \bar{z}^r\|^2 + \mathbb{E}\|z^{r+1} - z^r\|^2 + \sum_{i=1}^{N} \mathbb{E}\|1/N\nabla g_i(z^r) - 1/N\nabla g_i(y_i^{r-1})\|^2 \right).$$

These steps will be verified one by one shortly. But let us suppose that they all hold true. Below we show that linear convergence can be obtained.

Combining step 4 and step 2 we conclude that there exists $\sigma_3 > 0$ such that for all $r \geq \bar{r}$

$$\mathbb{E}[Q^{r+1} - \bar{v}] \leq \sigma_3 \left( \tau \|\mathbb{E}[\nabla_{1/\beta}\tilde{f}(z^{r-1})]\|^2 + \mathbb{E}\|z^{r+1} - z^r\|^2 + \sum_{i=1}^{N} \mathbb{E}\|1/N\nabla g_i(z^r) - 1/N\nabla g_i(y_i^{r-1})\|^2 \right).$$

Then if we bound $\|\mathbb{E}(G^r)\|^2$ using step 3, we can simply make a $\sigma_4 > 0$ such that

$$\mathbb{E}[Q^{r+1} - \bar{v}] \leq \sigma_4 \left( \mathbb{E}\|z^{r+1} - z^r\|^2 + \sum_{i=1}^{N} \mathbb{E}\|1/N\nabla g_i(z^r) - 1/N\nabla g_i(y_i^{r-1})\|^2 \right).$$

Finally, applying step 1 we reach the following bound for $\mathbb{E}[Q^{r+1} - \bar{v}]$:

$$\mathbb{E}[Q^{r+1} - \bar{v}] \leq \frac{\sigma_4}{\sigma_1} \mathbb{E}[Q^r - Q^{r+1}], \quad \forall\, r \geq \bar{r},$$

which further implies that for $\sigma_5 = \frac{\sigma_4}{\sigma_1} > 0$, we have

$$\mathbb{E}[\Delta^{r+1}] \leq \frac{\sigma_5}{1 + \sigma_5} \mathbb{E}[\Delta^r], \quad \forall\, r \geq \bar{r}.$$

Now let us verify the correctness of each step. Step 1 can be directly obtained from equation (2.12). Step 2 is exactly Lemma (5.1). Step 3 can be verified using a similar derivation as in (5.49)[5].

Below let us prove the step 4, which is a bit involved. From (2.7) we know that

$$z^{r+1} = \arg\min_z h(z) + g_0(z) + \sum_{i=1}^{N} \langle \lambda_i^r, x_i^{r+1} - z \rangle + \frac{\eta_i}{2}\|x_i^{r+1} - z\|^2.$$

This implies that

$$h(z^{r+1}) + g_0(z^{r+1}) + \sum_{i=1}^{N} \langle \lambda_i^r, x_i^{r+1} - z^{r+1} \rangle + \frac{\eta_i}{2}\|x_i^{r+1} - z^{r+1}\|^2$$

$$\leq h(\bar{z}^r) + g_0(\bar{z}^r) + \sum_{i=1}^{N} \langle \lambda_i^r, x_i^{r+1} - \bar{z}^r \rangle + \frac{\eta_i}{2}\|x_i^{r+1} - \bar{z}^r\|^2. \tag{5.66}$$

Rearranging the terms, we obtain

$$h(z^{r+1}) + g_0(z^{r+1}) - h(\bar{z}^r) - g_0(\bar{z}^r) \leq \sum_{i=1}^{N} \langle \lambda_i^r, z^{r+1} - \bar{z}^r \rangle + \frac{\eta_i}{2}\|x_i^{r+1} - \bar{z}^r\|^2.$$

Using this inequality we have:

$$Q^{r+1} - \bar{v} \le \sum_{i=1}^{N} 1/N \left( g_i(z^{r+1}) - g_i(\bar{z}^r) \right) + \langle \lambda_i^r, z^{r+1} - \bar{z}^r \rangle$$

$$+ \sum_{i=1}^{N} \frac{\eta_i}{2} \|x_i^{r+1} - \bar{z}^r\|^2 + \|1/N(\nabla g_i(z^r) - \nabla g_i(y_i^{r-1}))\|^2. \qquad (5.67)$$

The first term in RHS can be bounded as follows:

$$\sum_{i=1}^{N} 1/N \left( g_i(z^{r+1}) - g_i(\bar{z}^r) \right)$$

$$\overset{(a)}{\le} \sum_{i=1}^{N} 1/N \langle \nabla g_i(\bar{z}^r), z^{r+1} - \bar{z}^r \rangle + L_i/2N \|z^{r+1} - \bar{z}^r\|^2$$

$$\le \sum_{i=1}^{N} 1/N \langle \nabla g_i(\bar{z}^r) + \nabla g_i(z^{r+1}) - \nabla g_i(z^{r+1}), z^{r+1} - \bar{z}^r \rangle + L_i/2N \|z^{r+1} - \bar{z}^r\|^2$$

$$\overset{(b)}{\le} \sum_{i=1}^{N} 1/N \langle \nabla g_i(z^{r+1}), z^{r+1} - \bar{z}^r \rangle + 3L_i/2N \|z^{r+1} - \bar{z}^r\|^2,$$

where $(a)$ is true due to the descent lemma; and $(b)$ comes from the Lipschitz continuity of the $\nabla g_i$. Plugging the above bound into (5.67), we further have:

$$Q^{r+1} - \bar{v} \le \sum_{i=1}^{N} 1/N \langle \nabla g_i(z^{r+1}) - \nabla g_i(y_i^{r-1}), z^{r+1} - \bar{z}^r \rangle + 3L_i/2N \|z^{r+1} - \bar{z}^r\|^2$$

$$+ \frac{\eta_i}{2} \|x_i^{r+1} - \bar{z}^r\|^2 + \|1/N(\nabla g_i(z^r) - \nabla g_i(y_i^{r-1}))\|^2$$

$$= \sum_{i=1}^{N} 1/N \langle \nabla g_i(z^{r+1}) + \nabla g_i(z^r) - \nabla g_i(z^r) - \nabla g_i(y_i^{r-1}), z^{r+1} - \bar{z}^r \rangle$$

$$+ \frac{\eta_i}{2} \|x_i^{r+1} - \bar{z}^r\|^2 + \|1/N(\nabla g_i(z^r) - \nabla g_i(y_i^{r-1}))\|^2 + 3L_i/2N \|z^{r+1} - \bar{z}^r\|^2,$$

where in the first inequality we have used the fact that $\lambda_i^r = -\frac{1}{N} \nabla g_i(y_i^{r-1})$; cf. (5.25). Applying the Cauchy-Schwartz inequality we further have:

$$Q^{r+1} - \bar{v} \le \sum_{i=1}^{N} 1/2 \|1/N \left( \nabla g_i(z^{r+1}) + \nabla g_i(z^r) \right)\|^2 + 1/2 \|z^{r+1} - \bar{z}^r\|^2$$

$$+ \sum_{i=1}^{N} 1/2 \|1/N \left( \nabla g_i(z^r) - \nabla g_i(y_i^{r-1}) \right)\|^2 + 1/2 \|z^{r+1} - \bar{z}^r\|^2$$

$$+ \frac{\eta_i}{2} \|x_i^{r+1} - \bar{z}^r\|^2 + \|1/N(\nabla g_i(z^r) - \nabla g_i(y_i^{r-1}))\|^2 + 3L_i/2N \|z^{r+1} - \bar{z}^r\|^2$$

$$\le \sum_{i=1}^{N} \left[ \frac{L_i^2}{2N^2} \|z^{r+1} - z^r\|^2 + \frac{3}{2N^2} \|g_i(z^r) - \nabla g_i(y_i^{r-1})\|^2 + \frac{\eta_i}{2} \|x_i^{r+1} - \bar{z}^r\|^2 \right]$$

$$+ \left( 1 + 3L_i/2N \right) \|z^{r+1} - \bar{z}^r\|^2. \qquad (5.68)$$

Now let us bound $\sum_{i=1}^{N} \frac{\eta_i}{2} \|x_i^{r+1} - \bar{z}^r\|^2$ in the above inequality:

$$\sum_{i=1}^{N} \frac{\eta_i}{2} \|x_i^{r+1} - \bar{z}^r\|^2 = \sum_{i=1}^{N} \frac{\eta_i}{2} \|x_i^{r+1} - z^{r+1} + z^{r+1} - \bar{z}^r\|^2$$

$$\leq \sum_{i=1}^{N} \eta_i \|x_i^{r+1} - z^{r+1}\|^2 + \eta_i \|z^{r+1} - \bar{z}^r\|^2$$

$$= \sum_{i=1}^{N} \eta_i \|x_i^{r+1} - z^r + z^r - z^{r+1}\|^2 + \eta_i \|z^{r+1} - \bar{z}^r\|^2$$

$$\leq \sum_{i=1}^{N} 2\eta_i \|x_i^{r+1} - z^r\|^2 + 2\eta_i \|z^r - z^{r+1}\|^2 + \eta_i \|z^{r+1} - \bar{z}^r\|^2.$$

Using the fact that $x_i^{r+1} = z^r$ when $i \neq i_r$ we further have:

$$\sum_{i=1}^{N} \frac{\eta_i}{2} \|x_i^{r+1} - \bar{z}^r\|^2 \leq 2\eta_{i_r} \|x_{i_r}^{r+1} - z^r\|^2 + \sum_{i=1}^{N} 2\eta_i \|z^r - z^{r+1}\|^2 + \eta_i \|z^{r+1} - \bar{z}^r\|^2$$

$$= \frac{2}{\alpha_{i_r}^2 \eta_{i_r}} \|\lambda_{i_r} + 1/N \nabla g_{i_r}(z^r)\|^2 + \sum_{i=1}^{N} 2\eta_i \|z^r - z^{r+1}\|^2 + \eta_i \|z^{r+1} - \bar{z}^r\|^2$$

$$= \frac{2}{\alpha_{i_r}^2 \eta_{i_r} N^2} \|\nabla g_{i_r}(z^r) - \nabla g_{i_r}(y_{i_r}^{r-1})\|^2$$

$$+ \sum_{i=1}^{N} 2\eta_i \|z^r - z^{r+1}\|^2 + \eta_i \|z^{r+1} - z^r + z^r - \bar{z}^r\|^2$$

$$\leq \frac{2}{\alpha_{i_r}^2 \eta_{i_r} N^2} \|\nabla g_{i_r}(z^r) - \nabla g_{i_r}(y_{i_r}^{r-1})\|^2$$

$$+ \sum_{i=1}^{N} 4\eta_i \|z^r - z^{r+1}\|^2 + 2\eta_i \|z^r - \bar{z}^r\|^2. \qquad (5.69)$$

Take expectation on both sides of the above equation and set $p_i = \alpha_i$, we obtain:

$$\sum_{i=1}^{N} \frac{\eta_i}{2} \mathbb{E} \|x_i^{r+1} - \bar{z}^r\|^2 \leq \sum_{i=1}^{N} \frac{2}{\alpha_i \eta_i} \mathbb{E} \|\nabla g_i(z^r) - \nabla g_i(y_i^{r-1})\|^2$$

$$+ \sum_{i=1}^{N} 4\eta_i \mathbb{E} \|z^r - z^{r+1}\|^2 + 2\eta_i \mathbb{E} \|z^r - \bar{z}^r\|^2.$$

Combining equations (5.68) and (5.69), eventually one can find $\sigma_3 > 0$ such that

$$\mathbb{E}[Q^{r+1} - \bar{v}] \leq \sigma_3 \left( \mathbb{E} \|z^r - \bar{z}\|^2 + \mathbb{E} \|z^{r+1} - z^r\|^2 + \sum_{i=1}^{N} \mathbb{E} \|1/N \nabla g_i(z^r) - 1/N \nabla g_i(y_i^{r-1})\|^2 \right),$$

which completes the proof of Step 4.

In summary, we have shown that Step 1 - 4 all hold true. Therefore we have shown that the NESTT-G converges Q-linearly. **Q.E.D.**

## 5.5 Some Key Properties of NESTT-E

To facilitate the following derivation, in this section we collect some key properties of NESTT-E.

First, for $i = i_r$, using the optimality condition for $x_i$ update step (3.16) we have the following identity:

$$\frac{1}{N} \nabla g_{i_r}(x_{i_r}^{r+1}) + \lambda_{i_r}^r + \alpha_{i_r} \eta_{i_r}(x_{i_r}^{r+1} - z^{r+1}) = 0. \qquad (5.70)$$

Combined with the dual variable update step (3.17) we obtain

$$\frac{1}{N}\nabla g_{i_r}(x_{i_r}^{r+1}) = -\lambda_{i_r}^{r+1}. \tag{5.71}$$

Second, the optimality condition for the $z$-update is given by:

$$z^{r+1} = \operatorname{prox}_h \left[ z^{r+1} - \nabla_z(L(x^r, z, \lambda^r) - h(z)) \right] \tag{5.72}$$

$$= \operatorname{prox}_h \left[ z^{r+1} - \sum_{i=1}^{N} \eta_i \left( z^{r+1} - x_i^r - \frac{\lambda_i^r}{\eta_i} \right) - \nabla g_0(z^{r+1}) \right]. \tag{5.73}$$

## 5.6   Proof of Theorem 3.1

To prove this result, we need a few lemmas.

For notational simplicity, define new variables $\{\hat{x}_i^{r+1}\}$, $\{\hat{\lambda}_i^{r+1}\}$ by

$$\hat{x}_i^{r+1} := \arg\min_{x_i} U_i(x_i, z^{r+1}, \lambda_i^r), \quad \hat{\lambda}_i^{r+1} := \lambda_i^r + \alpha_i \eta_i \left( \hat{x}_i^{r+1} - z^{r+1} \right), \quad \forall i. \tag{5.74}$$

These variables are the *virtual variables* generated by updating all variables at iteration $r+1$. Also define:

$$L^r := L(x^r, z^r; \lambda^r), \quad w := (x, z, \lambda), \quad \beta := \frac{1}{\sum_{i=1}^{N} \eta_i}, \quad c_i := \frac{L_i^2}{\alpha_i \eta_i N^2} - \frac{\gamma_i}{2} + \frac{1-\alpha_i}{\alpha_i}\frac{L_i}{N}$$

First, we need the following lemma to show that the size of the successive difference of the dual variables can be upper bounded by that of the primal variables. This is a simple consequence of (5.71); also see [R2, Lemma 2.1]. We include the proof for completeness.

**Lemma 5.2.** *Suppose assumption A holds. Then for NESTT-E algorithm, the following are true:*

$$\|\lambda_i^{r+1} - \lambda_i^r\|^2 \le \frac{L_i^2}{N^2}\|x_i^{r+1} - x_i^r\|^2, \quad \|\hat{\lambda}_i^{r+1} - \lambda_i^r\|^2 \le \frac{L_i^2}{N^2}\|\hat{x}_i^{r+1} - x_i^r\|^2, \; \forall\, i. \tag{5.75a}$$

**Proof.** We only show the first inequality. The second one follows an analogous argument.

To prove (5.75a), first note that the case for $i \ne i_r$ is trivial, as both sides of (5.75a) are zero. For the index $i_r$, we have a closed-form expression for $\lambda_{i_r}^r$ following (5.71). Notice that for any given $i$, the primal-dual pair $(x_i, \lambda_i)$ is always updated at the same iteration. Therefore, if for each $i$ we choose the initial solutions in a way such that $\lambda_i^0 = -\nabla g_i(x_i^0)$, then we have

$$\frac{1}{N}\nabla g_i(x_i^{r+1}) = -\lambda_i^{r+1} \quad \forall\, i = 1, 2, \cdots N. \tag{5.76}$$

Combining (5.76) with Assumption A-(a) yields the following:

$$\|\lambda_i^{r+1} - \lambda_i^r\| = \frac{1}{N}\|\nabla g_i(x_i^{r+1}) - \nabla g_i(x_i^r)\| \le \frac{L_i}{N}\|x_i^{r+1} - x_i^r\|.$$

The proof is complete.                                                                **Q.E.D.**

Second, we bound the successive difference of the potential function.

**Lemma 5.3.** *Suppose Assumption A holds true. Then the following holds for NESTT-E*

$$\mathbb{E}[L^{r+1} - L^r | x^r, z^r] \le -\frac{\gamma_z}{2}\|z^{r+1} - z^r\|^2 + \sum_{i=1}^{N} p_i c_i \|x_i^r - \hat{x}_i^{r+1}\|^2. \tag{5.77}$$

**Proof.** First let us split $L^{r+1} - L^r$ in the following way:

$$L^{r+1} - L^r = L^{r+1} - L(x^{r+1}, z^{r+1}; \lambda^r) + L(x^{r+1}, z^{r+1}; \lambda^r) - L^r. \tag{5.78}$$

The first two terms in (5.78) can be bounded by

$$L^{r+1} - L(x^{r+1}, z^{r+1}; \lambda^r) = \sum_{i=1}^{N} \langle \lambda_i^{r+1} - \lambda_i^r, x_i^{r+1} - z^{r+1} \rangle$$

$$\overset{(a)}{=} \frac{1}{\alpha_{i_r} \eta_{i_r}} \| \lambda_{i_r}^{r+1} - \lambda_{i_r}^r \|^2 \overset{(b)}{\leq} \frac{L_{i_r}^2}{N^2 \alpha_{i_r} \eta_{i_r}} \| x_{i_r}^{r+1} - x_{i_r}^r \|^2 \tag{5.79}$$

where in (a) we have used (3.17), and the fact that $\lambda_i^{r+1} - \lambda_i^r = 0$ for all variable blocks except $i_r$th block; (b) is true because of Lemma 5.2.

The last two terms in (5.78) can be written in the following way:

$$L(\{x_i^{r+1}\}, z^{r+1}; \lambda^r) - L^r$$
$$= L(x^{r+1}, z^{r+1}; \lambda^r) - L(x^r, z^{r+1}; \lambda^r) + L(x^r, z^{r+1}; \lambda^r) - L^r. \tag{5.80}$$

The first two terms in (5.80) characterizes the change of the Augmented Lagrangian before and after the update of $x$. Note that $x$ updates do not directly optimize the augmented Lagrangian. Therefore the characterization of this step is a bit involved. We have the following:

$$L(x^{r+1}, z^{r+1}; \lambda^r) - L(x^r, z^{r+1}; \lambda^r)$$

$$\overset{(a)}{\leq} \sum_{i=1}^{N} \left( \langle \nabla_i L(x^{r+1}, z^{r+1}; \lambda^r), x_i^{r+1} - x_i^r \rangle - \frac{\gamma_i}{2} \| x_i^{r+1} - x_i^r \|^2 \right)$$

$$\overset{(b)}{=} \langle \nabla_{i_r} L(x^{r+1}, z^{r+1}; \lambda^r), x_{i_r}^{r+1} - x_{i_r}^r \rangle - \frac{\gamma_{i_r}}{2} \| x_{i_r}^{r+1} - x_{i_r}^r \|^2$$

$$\overset{(c)}{=} \langle \eta_{i_r} (1 - \alpha_{i_r})(x_{i_r}^{r+1} - z^{r+1}), x_{i_r}^{r+1} - x_{i_r}^r \rangle - \frac{\gamma_{i_r}}{2} \| x_{i_r}^{r+1} - x_{i_r}^r \|^2$$

$$\overset{(d)}{=} \left\langle \frac{1 - \alpha_{i_r}}{\alpha_{i_r}} (\lambda_{i_r}^{r+1} - \lambda_{i_r}^r), x_{i_r}^{r+1} - x_{i_r}^r \right\rangle - \frac{\gamma_{i_r}}{2} \| x_{i_r}^{r+1} - x_{i_r}^r \|^2$$

$$\leq \frac{1 - \alpha_{i_r}}{\alpha_{i_r}} \left( \frac{1}{2L_{i_r}/N} \| \lambda_{i_r}^{r+1} - \lambda_{i_r}^r \|^2 + \frac{L_{i_r}}{2N} \| x_{i_r}^{r+1} - x_{i_r}^r \|^2 \right) - \frac{\gamma_{i_r}}{2} \| x_{i_r}^{r+1} - x_{i_r}^r \|^2$$

$$\overset{(e)}{\leq} \frac{1 - \alpha_{i_r}}{\alpha_{i_r}} \frac{L_{i_r}}{N} \| x_{i_r}^{r+1} - x_{i_r}^r \|^2 - \frac{\gamma_{i_r}}{2} \| x_{i_r}^{r+1} - x_{i_r}^r \|^2 \tag{5.81}$$

where

- (a) is true because $L(x, z, \lambda)$ is strongly convex with respect to $x_i$.
- (b) is true because when $i \neq i_r$, we have $x_i^{r+1} = x_i^r$.
- (c) is true because $x_{i_r}^{r+1}$ is optimal solution for the problem $\min U_{i_r}(x_{i_r}, z^{r+1}, \lambda_{i_r}^r)$ (satisfying (5.70)), and we have used the optimality of such $x_{i_r}^{r+1}$.
- (d) and (e) are due to Lemma 5.2.

Similarly, the last two terms in (5.80) can be bounded using equation (5.70) and the strong convexity of function $L$ with respect to the variable $z$. Therefor We have:

$$L(x^r, z^{r+1}, \lambda^r) - L^r \leq -\frac{\gamma_z}{2} \| z^{r+1} - z^r \|^2. \tag{5.82}$$

Combining equations (5.79), (5.81) and (5.82), eventually we have:

$$L^{r+1} - L(x^r, z^{r+1}, \lambda^r) \leq c_{i_r} \| x_{i_r}^r - x_{i_r}^{r+1} \|^2 \tag{5.83}$$

$$L^{r+1} - L^r \leq -\frac{\gamma_z}{2} \| z^{r+1} - z^r \|^2 + c_{i_r} \| x_{i_r}^r - x_{i_r}^{r+1} \|^2 \tag{5.84}$$

Taking expectation on both side of this inequality with respect to $i_r$, we can conclude that:

$$\mathbb{E}[L^{r+1} - L^r \mid z^r, x^r] \leq -\frac{\gamma_z}{2} \| z^{r+1} - z^r \|^2 + \sum_{i=1}^{N} p_i c_i \| x_i^r - \hat{x}_i^{r+1} \|^2 \tag{5.85}$$

where $p_i$ is the probability of picking $i$th block. The lemma is proved. **Q.E.D.**

**Lemma 5.4.** *Suppose that Assumption A is satisfied, then $L^r \geq \underline{f}$.*

**Proof.** Using the definition of the augmented Lagrangian function we have:

$$L^{r+1} = \sum_{i=1}^{N} \left( \frac{1}{N} g_i(x_i^{r+1}) + \langle \lambda_i^{r+1}, x_i^{r+1} - z^{r+1} \rangle + \frac{\eta_i}{2} \| x_i^{r+1} - z^{r+1} \|^2 \right) + g_0(z^{r+1}) + p(z^{r+1})$$

$$\overset{(a)}{=} \sum_{i=1}^{N} \left( \frac{1}{N} g_i(x_i^{r+1}) + \frac{1}{N} \langle \nabla g_i(x_i^{r+1}), z^{r+1} - x_i^{r+1} \rangle + \frac{\eta_i}{2} \| x_i^{r+1} - z^{r+1} \|^2 \right) + g_0(z^{r+1}) + p(z^{r+1})$$

$$\overset{(b)}{\geq} \sum_{i=1}^{N} \frac{1}{N} g_i(z^{r+1}) + \left( \frac{\eta_i}{2} - \frac{L_i}{2N} \right) \| z^{r+1} - x_i^{r+1} \|^2 + g_0(z^{r+1}) + p(z^{r+1})$$

$$\overset{(c)}{\geq} \sum_{i=1}^{N} \frac{1}{N} g_i(z^{r+1}) + g_0(z^{r+1}) + p(z^{r+1}) \geq \underline{f} \tag{5.86}$$

where $(a)$ is true because of equation (5.71); $(b)$ follows Assumption A-(b); $(c)$ follows Assumption A-(d). The desired result is proven. **Q.E.D.**

**Proof of Theorem 3.1.** We first show that the algorithm converges to the set of stationary solutions, and then establish the convergence rate.

**Step 1. Convergence to Stationary Solutions**. Combining the descent estimate in Lemma 5.3 as well as the lower bounded condition in Lemma 5.4, we can again apply the Supermartigale Convergence Theorem [R1, Proposition 4.2] and conclude that

$$\| x_i^{r+1} - x_i^r \| \to 0, \quad \| z^{r+1} - z^r \| \to 0, \text{with probability 1.} \tag{5.87}$$

From Lemma 5.2 we have that the constraint violation is satisfied

$$\| \lambda^{r+1} - \lambda^r \| \to 0, \quad \| x_i^{r+1} - z^r \| \to 0. \tag{5.88}$$

The rest of the proof follows similar lines as in [R2, Theorem 2.4]. Due to space limitations we omit the proof.

**Step 2. Convergence Rate.** We first show that there exists a $\sigma_1(\alpha) > 0$ such that

$$\| \tilde{\nabla} L(w^r) \|^2 + \sum_{i=1}^{N} \frac{L_i^2}{N^2} \| x_i^r - z^r \|^2 \leq \sigma_1(\alpha) \left( \| z^r - z^{r+1} \|^2 + \sum_{i=1}^{N} \| x_i^r - \hat{x}_i^{r+1} \|^2 \right). \tag{5.89}$$

Using the definition of $\| \tilde{\nabla} L^r(w^r) \|$ we have:

$$\| \tilde{\nabla} L^r(w^r) \|^2 = \| z^r - \text{prox}_h \left[ z^r - \nabla_z (L^r - h(z^r)) \right] \|^2$$
$$+ \sum_{i=1}^{N} \left\| \frac{1}{N} \nabla g_i(x_i^r) + \lambda_i^r + \eta_i (x_i^r - z^r) \right\|^2. \tag{5.90}$$

From the optimality condition of the $z$ update (5.73) we have:

$$z^{r+1} = \text{prox}_h \left[ z^{r+1} - \sum_{i=1}^{N} \eta_i \left( z^{r+1} - x_i^r - \frac{\lambda_i^r}{\eta_i} \right) - \nabla g_0(z^{r+1}) \right].$$

Using this, the first term in equation (5.90) can be bounded as:

$$\|z^r - \text{prox}_h\left[z^r - \nabla_z(L^r - h(z^r))\right]\|$$

$$= \left\| z^r - z^{r+1} + z^{r+1} - \text{prox}_h\left[z^r - \sum_{i=1}^{N}\eta_i(z^r - x_i^r - \frac{\lambda_i^r}{\eta_i}) - \nabla g_0(z^r)\right]\right\|$$

$$\leq \|z^r - z^{r+1}\| + \left\| \text{prox}_h\left[z^{r+1} - \sum_{i=1}^{N}\eta_i\left(z^{r+1} - x_i^r - \frac{\lambda_i^r}{\eta_i}\right) - \nabla g_0(z^{r+1})\right]\right.$$

$$\left. - \text{prox}_h\left[z^r - \sum_{i=1}^{N}\eta_i(z^r - x_i^r - \frac{\lambda_i^r}{\eta_i}) - \nabla g_0(z^r)\right]\right\|$$

$$\leq 2\|z^{r+1} - z^r\| + \left(\sum_{i=1}^{N}\eta_i + L_0\right)\|z^r - z^{r+1}\|, \tag{5.91}$$

where in the last inequality we have used the nonexpansiveness of the proximity operator.

Similarly, the optimality condition of the $x_i$ subproblem is given by

$$\frac{1}{N}\nabla g_i(\hat{x}_i^{r+1}) + \lambda_i^r + \alpha_i\eta_i(\hat{x}_i^{r+1} - z^{r+1}) = 0. \tag{5.92}$$

Applying this identity, the second term in equation (5.90) can be written as follows:

$$\sum_{i=1}^{N}\left\|\frac{1}{N}\nabla g_i(x_i^r) + \lambda_i^r + \eta_i(x_i^r - z^r)\right\|^2$$

$$\overset{(a)}{=} \sum_{i=1}^{N}\left\|\frac{1}{N}\nabla g_i(x_i^r) - \frac{1}{N}\nabla g_i(\hat{x}_i^{r+1}) + \eta_i(x_i^r - z^r) - \alpha_i\eta_i(\hat{x}_i^{r+1} - z^{r+1})\right\|^2$$

$$= \sum_{i=1}^{N}\left\|\frac{1}{N}\nabla g_i(x_i^r) - \frac{1}{N}\nabla g_i(\hat{x}_i^{r+1}) + \eta_i(x_i^r - \hat{x}_i^{r+1} + \hat{x}_i^{r+1} - z^{r+1} + z^{r+1} - z^r) - \alpha_i\eta_i(\hat{x}_i^{r+1} - z^{r+1})\right\|^2$$

$$\overset{(b)}{\leq} 4\sum_{i=1}^{N}\left[\left(\frac{L_i^2}{N^2} + \eta_i^2 + \frac{(1-\alpha_i)^2 L_i^2}{N^2\alpha_i^2}\right)\|\hat{x}_i^{r+1} - x_i^r\|^2 + \eta_i^2\|z^{r+1} - z^r\|^2\right], \tag{5.93}$$

where (a) holds because of equation (5.92); (b) holds because of Lemma 5.2.

Finally, combining (5.91) and (5.93) leads to the following bound for proximal gradient

$$\|\tilde{\nabla}L^r\|^2 \leq \left(4\sum_{i=1}^{N}\eta_i^2 + \left(2 + L_0 + \sum_{i=1}^{N}\eta_i\right)^2\right)\|z^r - z^{r+1}\|^2$$

$$+ \sum_{i=1}^{N}4\left(\frac{L_i^2}{N^2} + \eta_i^2 + \frac{(1-\alpha_i)^2 L_i}{N^2\alpha_i^2}\right)\|x_i^r - \hat{x}_i^{r+1}\|^2. \tag{5.94}$$

Also note that:

$$\sum_{i=1}^{N}\frac{L_i^2}{N^2}\|x_i^r - z^r\|^2 \leq \sum_{i=1}^{N}3\frac{L_i^2}{N^2}\left[\|x_i^r - \hat{x}_i^{r+1}\|^2 + \|\hat{x}_i^{r+1} - z^{r+1}\|^2 + \|z^{r+1} - z^r\|^2\right]$$

$$= \sum_{i=1}^{N}3\frac{L_i^2}{N^2}\left[\|x_i^r - \hat{x}_i^{r+1}\|^2 + \frac{1}{\alpha_i^2\eta_i^2}\|\hat{\lambda}_i^{r+1} - \lambda_i^r\|^2 + \|z^{r+1} - z^r\|^2\right]$$

$$\leq \sum_{i=1}^{N}3\frac{L_i^2}{N^2}\left[\|x_i^r - \hat{x}_i^{r+1}\|^2 + \frac{L_i^2}{\alpha_i^2\eta_i^2 N^2}\|\hat{x}_i^{r+1} - x_i^r\|^2 + \|z^{r+1} - z^r\|^2\right].$$

$$\tag{5.95}$$

The two inequalities (5.94) – (5.95) imply that:

$$\|\tilde{\nabla}L^r\|^2 + \sum_{i=1}^{N} \frac{L_i^2}{N^2}\|x_i^r - z^r\|^2$$

$$\leq \left(\sum_{i=1}^{N} 4\eta_i^2 + (2 + \sum_{i=1}^{N}\eta_i + L_0)^2 + 3\sum_{i=1}^{N}\frac{L_i^2}{N^2}\right)\|z^r - z^{r+1}\|^2$$

$$+ \sum_{i=1}^{N}\left(4\left(\frac{L_i^2}{N^2} + \eta_i^2 + (\frac{1}{\alpha_i}-1)^2\frac{L_i^2}{N^2}\right) + 3\left(\frac{L_i^4}{\alpha_i N^4 \eta_i^2} + \frac{L_i^2}{N^2}\right)\right)\|x_i^r - \hat{x}_i^{r+1}\|^2. \qquad (5.96)$$

Define the following quantities:

$$\hat{\sigma}_1(\alpha) = \max_i \left\{4\left(\frac{L_i^2}{N^2} + \eta_i^2 + \left(\frac{1}{\alpha_i}-1\right)^2\frac{L_i^2}{N^2}\right) + 3\left(\frac{L_i^4}{\alpha_i\eta_i^2 N^4} + \frac{L_i^2}{N^2}\right)\right\}$$

$$\tilde{\sigma}_1 = \sum_{i=1}^{N} 4\eta_i^2 + (2 + \sum_{i=1}^{N}\eta_i + L_0)^2 + 3\sum_{i=1}^{N}\frac{L_i^2}{N^2}.$$

Setting $\sigma_1(\alpha) = \max(\hat{\sigma}_1(\alpha), \tilde{\sigma}_1) > 0$, we have

$$\|\tilde{\nabla}L^r\|^2 + \sum_{i=1}^{N}\frac{L_i^2}{N^2}\|x_i^r - z^r\|^2 \leq \sigma_1(\alpha)\left(\|z^r - z^{r+1}\|^2 + \sum_{i=1}^{N}\|x_i^r - \hat{x}_i^{r+1}\|^2\right). \qquad (5.97)$$

From Lemma 5.3 we know that

$$\mathbb{E}[L^{r+1} - L^r | z^r, x^r] \leq -\frac{\gamma_z}{2}\|z^{r+1} - z^r\|^2 + \sum_{i=1}^{N} p_i c_i \|x_i^r - \hat{x}_i^{r+1}\|^2 \qquad (5.98)$$

Note that $\gamma_z = \sum_{i=1}^{N}\eta_i - L_0$, then define $\hat{\sigma}_2$ and $\tilde{\sigma}_2$ as

$$\hat{\sigma}_2(\alpha) = \max_i\left\{p_i\left(\frac{\gamma_i}{2} - \frac{L_i^2}{\alpha_i\eta_i N^2} - \frac{1-\alpha_i}{\alpha_i}\frac{L_i}{N}\right)\right\}$$

$$\tilde{\sigma}_2 = \frac{\sum_{i=1}^{N}\eta_i - L_0}{2}.$$

We can set $\sigma_2(\alpha) = \max(\hat{\sigma}_2(\alpha), \tilde{\sigma}_2)$ to obtain

$$E[L^r - L^{r+1}|x^r, z^r] \geq \sigma_2(\alpha)\left(\sum_{i=1}^{N}\|\hat{x}_i^{r+1} - x_i^r\|^2 + \|z^{r+1} - z^r\|^2\right). \qquad (5.99)$$

Combining (5.97) and (5.99) we have

$$H(w^r) = \|\tilde{\nabla}L^r\|^2 + \sum_{i=1}^{N} L_i^2/N\|x_i^r - z^r\|^2 \leq \frac{\sigma_1(\alpha)}{\sigma_2(\alpha)}E[L^r - L^{r+1}|F^r].$$

Let us set $C(\alpha) = \frac{\sigma_1(\alpha)}{\sigma_2(\alpha)}$ and take expectation on both side of the above equation to obtain:

$$\mathbb{E}[H(w^r)] \leq C(\alpha)E[L^r - L^{r+1}]. \qquad (5.100)$$

Summing both sides of the above inequality over $r = 1, \cdots, R$, we obtain:

$$\sum_{r=1}^{R}\mathbb{E}[H(w^r)] \leq C(\alpha)E[L^1 - L^{R+1}]. \qquad (5.101)$$

Using the definition of $w^m = (x^m, z^m, \lambda^m)$, and following the same line of argument as Theorem (2.1) we eventually conclude that

$$\mathbb{E}[H(w^m)] \leq \frac{C(\alpha)\mathbb{E}[L^1 - L^{R+1}]}{R}. \qquad (5.102)$$

The proof is complete. **Q.E.D.**

## 5.7 Proof of Proposition 4.1

Applying the optimality condition on $z$ subproblem in (5.32) we have:

$$z^{r+1} = \underset{z}{\operatorname{argmin}}\, h(z) + g_0(z) + \frac{\beta}{2}\|z - u^{r+1}\|^2 \tag{5.103}$$

where the variable $u^{r+1}$ is given by (cf. (5.29))

$$u^{r+1} = \beta \sum_{i=1}^{N} (\lambda_i^r + \eta_i x_i^{r+1}). \tag{5.104}$$

Now from one of the key properties of NESTT-G [cf. Section 5.1, equation (5.28)], we have that

$$u^{r+1} = z^r - \beta \left( \frac{1}{N\alpha_{i_r}} \left( \nabla g_{i_r}(z^r) - \nabla g_{i_r}(y_{i_r}^{r-1}) \right) + \frac{1}{N}\sum_{i=1}^{N} \nabla g_i(y_i^{r-1})N \right). \tag{5.105}$$

This verifies the claim. **Q.E.D.**

## 5.8 References

[R1] D. P. Bertsekas and J. N. Tsitsiklis. Neuro-Dynamic Programming. Athena Scientific, Belmont, MA, 1996.

[R2] M. Hong, Z.-Q. Luo, and M. Razaviyayn. Convergence analysis of alternating direction method of multipliers for a family of nonconvex problems. SIAM Journal On Optimization, 26(1):337 - 364, 2016

[R3] P. Tseng and S. Yun. A coordinate gradient descent method for nonsmooth separable minimization. Mathematical Programming, 117:387 - 423, 2009.

[R4] Z.-Q. Luo and P. Tseng. Error bounds and the convergence analysis of matrix splitting algorithms for the affine variational inequality problem. SIAM Journal on Optimization, pages 43 - 54, 1992.

[R5] Z.-Q. Luo and P. Tseng. On the linear convergence of descent methods for convex essentially smooth minimization. SIAM Journal on Control and Optimization, 30(2):408 - 425, 1992.

## Footnotes

[5]We simply need to replace $-z^{r-1} + \text{prox}_h^{1/\beta}[u^{r-1} - \beta\nabla g_0(z^{r-1})]$ in step (a) of (5.49) by $-z^r + \text{prox}_h^{1/\beta}[u^r - \beta\nabla g_0(z^r)]$ and using the same derivation.