[Reviews · NeurIPS 2016]

Reviewer 1

Summary

The paper presents a primal-dual type algorithm to solve non-convex composite optimization problems and establish its rate of convergence. The algorithm targets for solving a finite sum optimization problem.

Qualitative Assessment

The author provides a framework to generalize randomized incremental gradient methods for the nonconvex setting. Here are some suggestions to improve the paper: 1)The authors should be more careful when stating that their algorithm can be O(N) times better than deterministic algorithms. The complexity of deterministic algorithms depends on the Lipschitz constant of the function $\sum_i f_i(x)$, which can be much smaller than $\sum_i L_i$. 2) The authors are suggested to cite Ghadimi, Lan and Zhang (Math Prog), which discusses non-convex composite optimization (both deterministic and stochastic). 3) It seems to the referee that if a mini-batch approach is used (similar to Ghadimi, Lan and Zhang) with the batch size depending on the variance (now the number of terms), the authors might be able to gain an O(1/N^{1/3}) factor as discussed in Section 4.

Confidence in this Review

2-Confident (read it all; understood it all reasonably well)


Reviewer 2

Summary

The paper focuses on non-convex problems with objectives consisting of a finite sum of smooth nonconvex loss functions and a nonsmooth convex regularization. In contrast to the recent incremental algorithms, the authors propose new randomized variants of (linearized) ADMM algorithm (or called primal-dual splitting by the authors) to solve the nonconvex problem, which converge sublinearly to stationary solution. Compared to batch gradient descent, the proposed algorithm achieves better complexity of gradient evaluations in certain cases (e.g. when Lipschitz constants are unbalanced). Some interesting connection to SAG/SAGA is revealed.

Qualitative Assessment

The idea of addressing finite-sum problems through augmented Lagrangian and ADMM type algorithm is straightforward but new. Convergence results of the new randomized variation for non-convex problems are also novel, to my best knowledge. Despite the notable originality, I think a couple of places need to be further discussed or clarified. 1. Although existing work on nonconvex SVRG/SAGA algorithms [1,21] only provide analysis for smooth problems, I would presume similar results (replacing gradient by prox-gradient) can also be derived with slight modification given that the original algorithms are designed to handle nonsmooth regularization and adapted to non-uniform sampling as well. So it would be good if the author could add such comparison to nonconvex SVRG/SAGA in Table 1 when applicable. 2. It appears to me that in the NESTT-G algorithm (without reducing to the single variable form) , at each iteration, setting the remaining (N-1) x variables to z would require O(Nd) computation and memory cost, which is not negligible. In some sense, the total computation cost could be O(N) times larger than the number of gradient evaluations. Hence, the comparison between NESTT-G and GD is not quite fair. 3. When reducing to the single variable form, the NESTT-G algorithm reduces to SAGA with special choice of stepsize. However, with uniform Lipschitz constants, the complexity of NESTT-G is worse than SAGA, Perhaps the author could explain what reason leads to the gap, and whether this can be remedied. 4. In the experiment, comparison to Gradient Descent (and non-uniform SAGA/SVRG) should be added. It would be more interesting to see results with larger size instead of just 500 data points. The notation of uniform sampling is confusing and inconsistent with the theory. 5. Minor issues: wrong relation in Thm 2.2; typo in (3.17). == post-rebuttal answer== I have read the author's rebuttal and updated my scores accordingly.

Confidence in this Review

2-Confident (read it all; understood it all reasonably well)


Reviewer 3

Summary

This paper gives insights on relations between SAGA and primal-dual methods. The goal is to generalize stochastic gradient techniques to non-convex functions and splitting methods.

Qualitative Assessment

I think that the paper is interesting to read and brings new ideas. However, the proof should be corrected and in particular the conditional expectations (see fatal flaw). Other remarks. Line 47: "NESTT converges sublinearly to stationary solution set". The algorithm converges to a point that belongs to the stationary solution set, no to the set itself. Line 50: optimiztiaon -> optimization Line 54: uniform -> equal Line 64: th -> the Line 110: the only source of randomness in the method is i(r) so it would be more natural to define the sigma-field from i(r) than from x^r and z^r. Line 111, (2.8a): a gradient sign is missing Line 111, (2.8b): as (2.8b) uses (2.8c), the two equalities should be swapped. Line 114: What is the descent lemma? Line 115: Where does the 3 come from? Line 124: G^m -> G^R Line 171, (3.13): It would be more consistent with NESTT-G to define z^{r+1} from x^{r+1} and lambda^{r+1}. Line 180: the last semicolon is in the wrong place. Figure 1: Talking of optimality gap is not correct in your setting. This is rather a stationarity gap. Line 240: R stands for the number of iteration and for the parameter that controls sparsity. Line 334: the argmin defines the prox of h+g_0, not the prox of h. (same thing in Eq (5.71)) Line 342: You do not need to talk about the gradient of g_0 since you are computing the prox of h+g_0. Then, you can replace the 3 in the fifth line by a 0.

Confidence in this Review

3-Expert (read the paper in detail, know the area, quite certain of my opinion)


Reviewer 4

Summary

The paper proposes two stochastic algorithms for solving a non convex and non smooth composite optimization problem. The objective function is the sum of a finite number of smooth functions (possibly nonconvex) and a convex function (possibly nonsmooth). The proposed algorithms are respectively: NESTT-G and NESTT-E. NESTT-G requires a stochastic gradient step plus a gradient ascent on the dual variable, and a proximal step at each iteration. NESTT-E belongs to the class of stochastic ADMM algorithms. The main contribution is a global convergence result to a critical point with a convergence rate, that can be linear under additional assumptions. Experimental results on a small scale non convex non smooth optimization problem are presented.

Qualitative Assessment

The paper addresses an interesting problem of extending ADMM-type algorithms to the non convex setting and provides theoretical guarantees. The numerical results looks encouraging, even if the considered problem in the experiments is small scale, and therefore not appropriate to evaluate the real impact of the proposed method. The proofs techniques used in the paper look quite standard, but in my opinion the contribution is significant. There are two weak points in the analysis. First, at each iteration, the minimization of the augmented lagrangian can be nontrivial. Indeed, it requires the computation of the proximity operator of the function g_0+h, and it is well-known that computing the prox of a sum is difficult, and not efficient in general. Second, the convergence result in Theorem 2.1 is very weak, since it establishes convergence only of a subsequence, and only in the case where a limit point exists. But the existence of a limit point is guaranteed only under additional assumptions such as coerciveness. The obtained convergence rate is for the residual, and not for the function values, nor for the distance to a critical point. Summarising, I think that the paper is an interesting contribution in

Confidence in this Review

3-Expert (read the paper in detail, know the area, quite certain of my opinion)


Reviewer 5

Summary

This paper proposes two distributed stochastic primal-dual splitting algorithms for solving nonconvex composite minimization problems. The authors assume that the first function can be decomposed into N+1 Lipschitz gradient functions, while the second term is convex. The main idea is to duplicate the variables and then using augmented Lagrangian framework. The authors design a distributed stochastic algorithm that allows one to solve the augmented Lagrangian subproblems in a distributed manner. They first prove a global convergence of the algorithm to a stationary point using non-uniform probability distribution. They also provide a local linear convergence under an error bound type condition. Then, the authors suggest a new variant of their algorithm called NESTT-E, that uses exact minimization. They also prove the convergence of this algorithm and compare to existing methods. A numerical example on a nonconvex quadratic problem over the l-1 norm is given to illustrate the theoretical results.

Qualitative Assessment

The reviewer thinks that this paper presents new results for the non-uniform probability distribution in solving nonconvex composite minimization problems. The algorithms are new and the convergence results are provided. As shown by the authors, their theoretical result has a significant improvement over existing results when using non-uniform distribution. However, the reviewer was not able to check the proofs in detail. In addition, the reviewer is not sure about the complexity per iteration, which is whether or not significantly increased when using non-uniform distribution. Overall, the paper is good and is deserved for NIPS.

Confidence in this Review

1-Less confident (might not have understood significant parts)